Ecological and Evolutionary Science

# Press Disturbance Alters Community Structure and Assembly Mechanisms of Bacterial Taxa and Functional Genes in Mesocosm-Scale Bioreactors

Ezequiel Santillan,[a,b] Florentin Constancias,[a] Stefan Wuertz[a,b,c]

[a]Singapore Centre for Environmental Life Sciences Engineering, Nanyang Technological University, Singapore
[b]Department of Civil and Environmental Engineering, University of California, Davis, California, USA
[c]School of Civil and Environmental Engineering, Nanyang Technological University, Singapore

**ABSTRACT** Press disturbances are of interest in microbial ecology, as they can drive microbial communities to alternative stable states. However, the effect of press disturbances in community assembly mechanisms, particularly with regard to taxa and functional genes at different levels of abundance (i.e., common and rare), remains largely unknown. Here, we tested the effect of a continuous alteration in substrate feeding scheme on the structure, function, and assembly of bacterial communities. Two sets of replicate 5-liter sequencing batch reactors were operated at two different organic carbon loads for a period of 74 days, following 53 days of acclimation after inoculation with sludge from a full-scale treatment plant. Temporal dynamics of community taxonomic and functional gene structure were derived from metagenomics and 16S rRNA gene metabarcoding data. Disturbed reactors exhibited different community function, structure, and assembly compared to undisturbed reactors. Bacterial taxa and functional genes showed dissimilar $\alpha$-diversity and community assembly patterns. Deterministic assembly mechanisms were generally stronger in disturbed reactors and in common fractions compared to rare ones. Function quickly recovered after the disturbance was removed, but community structure did not. Our results highlight that functional gene data from metagenomics can indicate patterns of community assembly that differ from those obtained from taxon data. This study reveals how a joint evaluation of assembly mechanisms and community structure of bacterial taxa and functional genes as well as ecosystem function can unravel the response of complex microbial systems to a press disturbance.

**IMPORTANCE** Ecosystem management must be viewed in the context of increasing frequencies and magnitudes of various disturbances that occur at different scales. This work provides a glimpse of the changes in assembly mechanisms found in microbial communities exposed to sustained changes in their environment. These mechanisms, deterministic or stochastic, can cause communities to reach a similar or variable composition and function. For a comprehensive view, we use a joint evaluation of temporal dynamics in assembly mechanisms and community structure for both bacterial taxa and their functional genes at different abundance levels, in both disturbed and undisturbed states. We further reverted the disturbance state to contrast recovery of function with community structure. Our findings are relevant, as very few studies have employed such an approach, while there is a need to assess the relative importance of assembly mechanisms for microbial communities across different spatial and temporal scales, environmental gradients, and types of disturbance.

**KEYWORDS** diversity, disturbance, community structure, stochastic assembly, deterministic assembly, abundant, null-model analysis, rare

Address correspondence to Stefan Wuertz, swuertz@ntu.edu.sg.

This work presents a joint evaluation of temporal dynamics in assembly mechanisms and community structure for both bacterial taxa and their functional genes at different abundance levels, at both disturbed and undisturbed states.

Microbes drive all biogeochemical cycles on Earth, with microbial communities providing important ecosystem functions that impact all other forms of life (1). Community structure, often described in terms of $\alpha$- and $\beta$-diversity, is thought to have an effect on ecosystem function (2). However, our capacity to predict and manage the functions of microbial communities and how they are linked to community structure is still limited (3). In this regard, engineered systems like sludge bioreactors for wastewater treatment constitute model systems for microbial ecology studies (4), with measurable ecosystem functions, such as carbon and ammonia removal, that not only are important in practice (5) but also involve complex microbial communities in a controlled environment (6).

In ecology, disturbances are believed to have direct effects on ecosystems by altering community structure and function (7). Press disturbances that impose a long-term continuous change of species abundances by altering the environment (8) are of interest in microbial ecology, as they can drive systems to alternative stable states with different community function and structure (9). These disturbances could occur in the form of environment modifications that are not directly harmful to organisms, while still providing less abundant community members opportunities to grow (10). In sludge bioreactors, a continuous alteration in the substrate feeding scheme can trigger changes in community function and structure, yet whether these changes are reproducible (11) and whether they can be reversed when the disturbance ceases remain unknown.

Community assembly mechanisms are inherently linked with ecosystem function, as they play an important role in shaping community structure (12). These mechanisms can be either deterministic (13) or stochastic (14), and they may act in combination to shape patterns of community assembly (15–18). Disturbance is thought to be a main factor driving these underlying mechanisms of community assembly (19), yet a predictive understanding of its effects is missing (20). Disturbance can promote stochastic assembly mechanisms that lead communities to divergent states of structure and function (21, 22); therefore, studies assessing its effects require replicated designs (23). Also, microbial communities within wastewater treatment systems have been shown to harbor a core group of 100 to 800 abundant (i.e., common) operational taxonomic units (OTUs) across plants and countries (24, 25). Activated sludge community dynamics have been suggested to differ for common and less abundant (i.e., rare) taxa on the grounds of distance-based analyses (26–29), proposing that common OTUs are driven by deterministic mechanisms of assembly. The contribution of the remaining rare taxa to biochemical transformations is as yet unknown, but their potential as a seed bank for gene and taxonomic diversity (30) merits consideration. Indeed, rare taxa and genes may become very abundant under conditions (31) that can be elicited by disturbance.

The contribution of assembly mechanisms is often quantified via null-model analyses (32). In studies of microbial systems, such analyses are usually focused on community data obtained from a fragment of the 16S rRNA gene (17, 18, 33–38), which is known to be highly conserved between different species of prokaryotes (39). On the other hand, the use of metagenomics would allow inference of assembly mechanisms from the whole-community DNA (40), enabling the assessment of variability in genes that are less conserved and that could unveil important aspects of community assembly. However, studies usually employ these sequencing methods separately, as their results are thought to be hard to reconcile (41). Microbial communities can also be assessed in terms of their functional gene structure (42), and while functional gene assembly can be assessed via microarray (22, 43) or clone library (44) approaches, a more complete view of the functional gene structure of the microbial community can be obtained through shotgun metagenomics. Hence, an assessment of community assembly mechanisms combining both 16S rRNA gene amplicon and shotgun metagenomics sequencing would provide important insights toward a better understanding of the effect of disturbance in community assembly, structure, and function, while still allowing a comparison with prior findings. Further, such evaluation is seldom carried out in complex and engineered microbial systems like bioreactors for wastewater treatment.

The objective of this work was to test the effect of a press disturbance by doubling

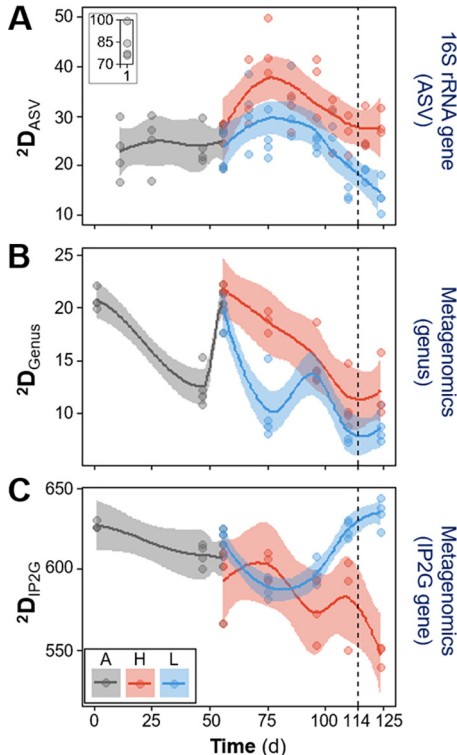

**FIG 1** Temporal dynamics of second-order true $\alpha$-diversity ($^2$D) for bacterial taxa and functional genes. (A) 16S rRNA gene sequencing at the ASV level; (B) metagenomics sequencing at the genus level; (C) metagenomics sequencing at the IP2G lowest gene level. Each point represents a different reactor for a given day. Phases: A, acclimation (gray; $n = 4$); L, low organic loading (blue; $n = 4$); H, high organic loading (red; $n = 3$). Vertical dashed lines indicate the shift from high to low organic loading. Lines display polynomial regression fitting, while shaded areas represent 95% confidence intervals. The inset (A) shows data points at day 1.

the organic load in a replicated set of activated sludge bioreactors at mesocosm scale. Based on our findings in a prior study at microcosm scale (21), we expected the disturbance to affect community function, structure, and assembly, with the hypothesis of a stronger deterministic effect at the disturbed level. Samples were analyzed using shotgun metagenomics, 16S rRNA gene metabarcoding, and effluent chemical characterization. Patterns of $\alpha$- and $\beta$-diversity were employed to assess temporal dynamics of community structure. Assembly mechanisms were evaluated through a mathematical null model on the effective bacterial turnover expressed as a proportion of total bacterial diversity. Finally, the disturbance was removed during the last 14 days of the study to evaluate if community function, structure, and assembly would display signs of recovery.

## RESULTS

**Dynamics of bacterial community structure.** Throughout the acclimation phase there was a significant change in bacterial community structure ($P$ determined by permutational analysis of variance [$P_{\text{PERMANOVA}}$] < 0.005) (Table S2) from the starting wastewater treatment plant (WWTP) inoculum to the acclimated sludge that was later distributed across reactors when the disturbance phase started, in terms of both $\alpha$-diversity (Fig. 1A) and $\beta$-diversity (Fig. 2A).

Patterns of $\alpha$-diversity varied with time and differed among treatments (Fig. 1; Fig. S1). Disturbed reactors displayed higher $\alpha$-diversity ($^2$D) for taxonomic data sets of both sequencing techniques employed (Fig. 1A and B). In contrast, undisturbed reactors displayed the highest $\alpha$-diversity of functional genes at the end of the disturbance phase (Fig. 1C). Lower-order compound $\alpha$-diversity ($^1$D) had similar patterns, while

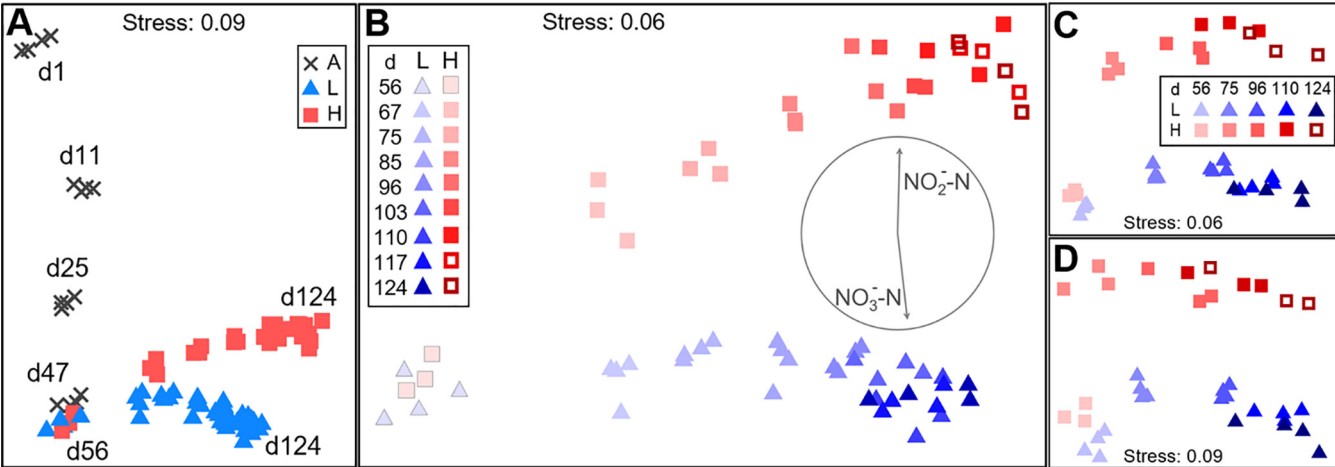

**FIG 2** Community structure dynamics for bacterial taxa and functional genes evaluated through NMDS ordination (Bray-Curtis $\beta$-diversity). (A and B) 16S rRNA gene sequencing at the ASV level; (C) metagenomics sequencing at the genus level; (D) metagenomics sequencing at the IP2G lowest gene level (same legend as C). Phases: A, acclimation (gray crosses, $n = 4$); L, low organic loading (blue triangles; $n = 4$); H, high organic loading (red squares; $n = 3$). Open red squares indicate the shift from high to low organic loading. Each point represents a different reactor on a given day. Days are indicated with text within the panel for days 1 to 124 (A) and with decreasing brightness from days 56 to 124 (B to D). Panel B includes Pearson's correlation vectors of nitrite and nitrate in the effluent.

richness ($^0D$) did not show any differences among treatments (Fig. S1). After the shift from high to low organic loading in disturbed reactors, treatments continued to vary in terms of $\alpha$-diversity, with $^2D$ being significantly different for both the amplicon sequence variant (ASV) (*P* at day 124 [$P_{d124}$] = 0.0046) and the InterPro-to-gene ontologies (IP2G) gene data sets ($P_{d124}$ = 0.0003) based on Welch's ANOVA.

During the disturbance phase, temporal patterns of $\beta$-diversity showed bacterial communities clustering separately for low- and high-organic-loading treatments, regardless of the sequencing method employed for both taxa and genes (Fig. 2). As per PERMANOVA, such differentiation was statistically significant from day 56 onwards for all data sets, with nonsignificant permutational analysis of dispersion (PERMDISP) results supporting the absence of heteroscedasticity (Table S2). Dissimilarity increased with time among replicates of both treatments across all data sets (Fig. 2B to D). There was no significant effect of the shift from high to low organic loading with regard to $\beta$-diversity.

Similar to observed diversity patterns, succession was evident after analysis of relative abundances of specific bacterial taxa. The 15 most abundant genera, assessed through 16S rRNA gene metabarcoding at the genus level, were different at each phase of the study (Fig. 3). Only seven taxa were present in the top 15 genera throughout the study, and their relative abundances differed with experimental phase. Succession was also observed in metagenomics data, at both the taxonomic bacterial genus level (Fig. S2, Text S1) and the functional gene level (Fig. 4; Fig. S3).

**Dynamics of ecosystem functions.** Ecosystem function dynamics were described in detail elsewhere (45). In short, there was a clear distinction between reactors receiving low and high organic loading, with the disturbed reactors displaying partial nitrification with high nitrite concentrations in the effluent (Fig. S4; Fig. 2B). Following the shift from high to low organic loading, a transition toward recovery of the nitrite oxidation function was observed, with variability across replicate reactors that resulted in only two of them recovering these functions after 14 days (Fig. 5).

**Dynamics of bacterial community assembly mechanisms.** A null-model approach was applied to quantify assembly mechanisms, based on the deviation of observed $\beta$-diversity from the expected $\beta$-diversity. Deterministic strength (DS), a metric of deterministic assembly, was generally higher for disturbed reactors regardless of the sequencing method (Fig. 6), with a higher separation for taxa (Fig. 6A and D) than for genes (Fig. 6G). DS decreased with time for metagenomics-based taxa and genes after acclimation (Fig. 6D to I), coinciding with a temporal increase in $\beta$-diversity within and

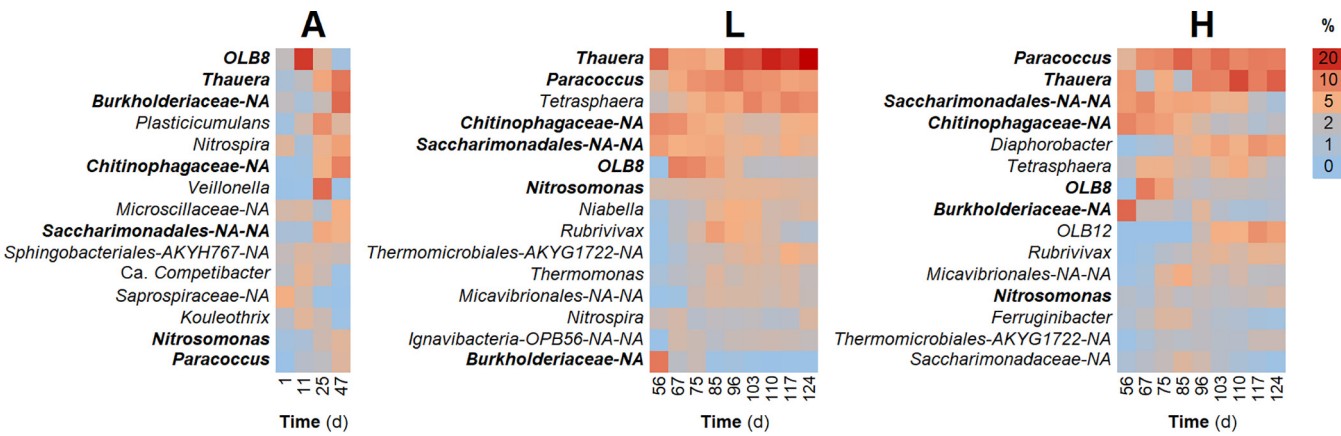

**FIG 3** Community structure dynamics for abundant bacterial taxa, assessed through 16S rRNA gene metabarcoding at the genus level. The 15 most abundant genera for each phase are shown. Genera that belong to the top 15 during all phases are in bold. Columns represent the average among reactors for a given phase and day. Phases: A, acclimation ($n = 4$); L, low organic loading ($n = 4$); H, high organic loading ($n = 3$).

among treatments (Fig. 2). Deterministic strength values were mostly above 50% for ASV data, indicating higher determinism (Fig. 6A). In contrast, most of the DS values for the metagenomics IP2G gene level data set were below 50%, marking stronger stochasticity (Fig. 6G). Further partition of the data sets into common (up to 90% accumulated reads) and rare (<10% accumulated reads) fractions showed that the common fraction (Fig. 6B, E, and H) generally had a higher DS than the rare one (Fig. 6C, F, and I). After the shift from high to low organic loading, similar DS values for both levels were observed only for the metagenomics gene data (Fig. 6G). Observed $\beta$-diversity was greater than expected for bacterial genera regardless of the sequencing method ($\overline{\beta_{exp}:\beta_{obs}} < 1$), while an opposite pattern was seen for functional genes ($\overline{\beta_{exp}:\beta_{obs}} > 1$) (Fig. 7). Detailed information on the parameters and outputs of the model is given in Table S3.

An additional null-model approach that assesses phylogenetic turnover using the $\beta$-nearest taxon index ($\beta$NTI) was applied to the ASV data. Both disturbed and undisturbed reactors had mean values of $\beta$NTI below $-2$, indicating deterministic assembly (Fig. 8A). This was also the case when only the common fraction was considered

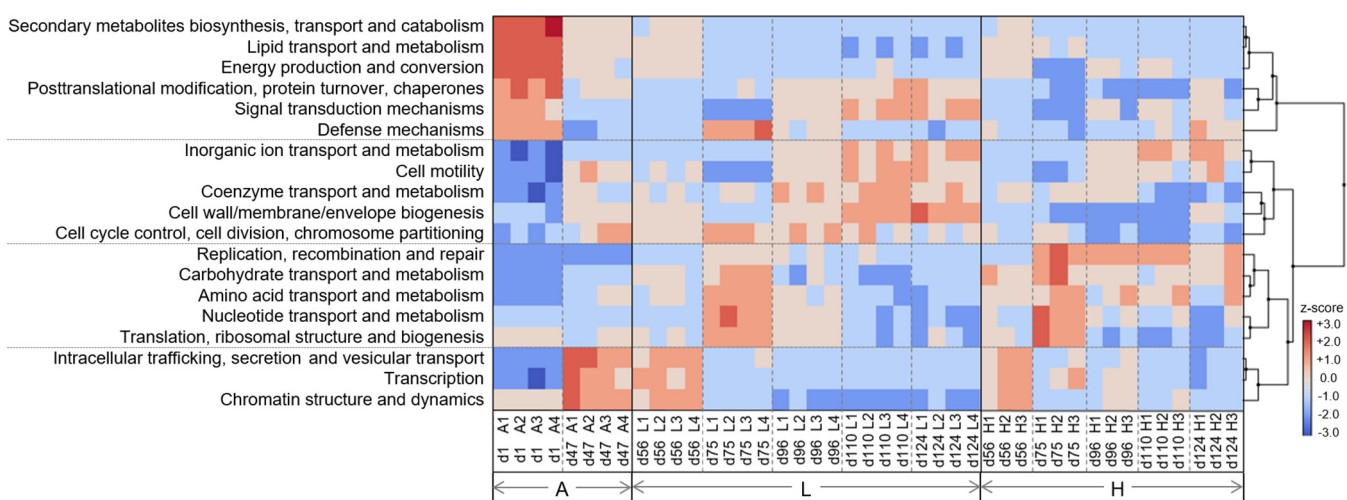

**FIG 4** Successional clusters of the 19 most abundant functional gene categories from the COG database (>10,000 reads) across reactors over time. Z-scores indicate how many standard deviations of the mean each sample contains (assigned reads per gene category, across all samples). Each column legend presents day number and reactor. Phases: A, acclimation ($n = 4$); L, low organic loading ($n = 4$); H, high organic loading ($n = 3$). Rectangles highlight groups of functional gene categories prevailing at different phases. Dashed vertical lines separate time points within the same phase. Dashed horizontal lines separate the four biggest clusters of trait complexes.

**FIG 5** Temporal effluent concentrations (in milligrams per liter) at the end of a reactor cycle, adapted from reference 45 to display bioreactor function during the last 14 days of the study. (A) Organic carbon as soluble chemical oxygen demand (COD); (B and C) nitrite (B) and nitrate (C) as nitrogen. Legend: H, high organic loading (red), where each symbol represents a different replicate reactor (H1, H2, and H3); L, low organic loading (blue), displaying average values to facilitate comparison ($n = 4$). Vertical dashed lines indicate the shift from high to low organic loading. Effluent profiles for the whole study are available in Fig. S4.

(Fig. 8B), whereas for the rare fraction, a |$\beta$NTI| of <2 indicated a significant effect of stochastic assembly (Fig. 8C). The observed $\beta$NTI patterns were not driven by a particular reactor (Fig. S5A to C). Phylogenetic Mantel correlogram analysis showed a significant phylogenetic signal but only across relatively short phylogenetic distances (Fig. S5D), supporting the use of $\beta$NTI.

## DISCUSSION

**Disturbance leads to different community function and structure.** Community function and structure, in terms of $\alpha$- and $\beta$-diversity, were clearly different in reactors

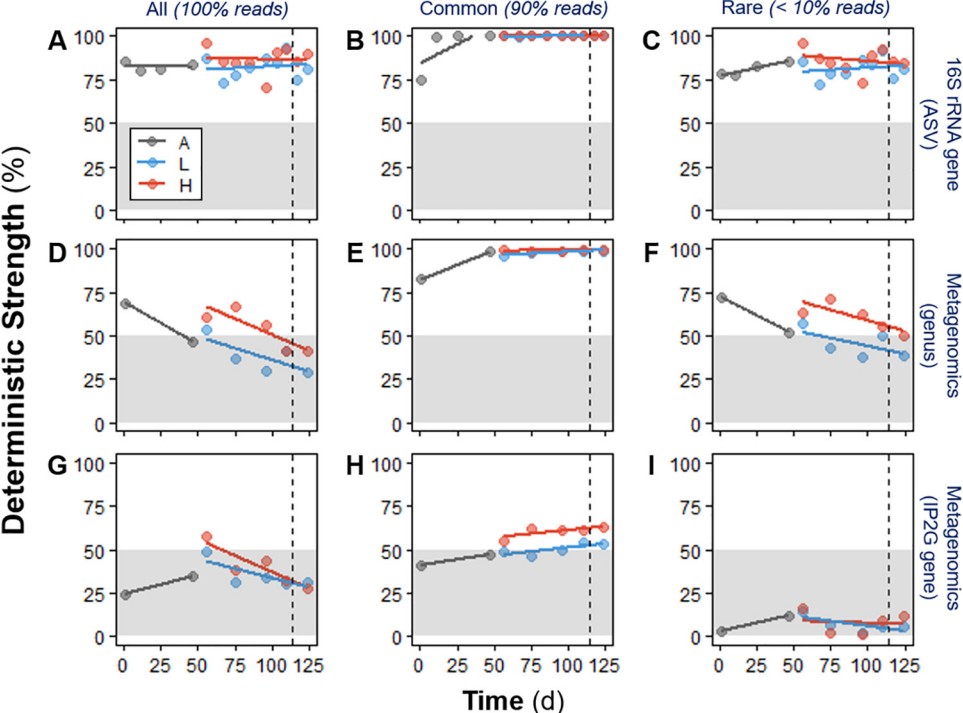

**FIG 6** Deterministic strength (DS) temporal dynamics for bacterial taxa and functional genes, derived from null-model analysis. (A to C) 16S rRNA gene sequencing at the ASV level; (D to F) metagenomics sequencing at the genus level; (G to I) metagenomics sequencing at the IP2G lowest gene level. Values were calculated using all taxa/genes (left panels [A, D, and G]), common taxa/genes (90% accumulated reads) (middle panels [B, E, and H]), and rare taxa/genes (<10% accumulated reads) (right panels [C, F, and I]). Phases: A, acclimation (gray); L, low organic loading (blue); H, high organic loading (red). Each point calculation involved all replicates ($n = 4$ for phases A and L; $n = 3$ for phase H) of each phase evaluated. Lines represent linear regression fitting. Vertical dashed lines indicate the shift from high to low organic loading. Zones of higher stochastic intensity within the panels are shaded.

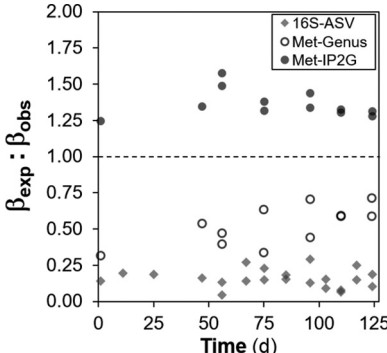

**FIG 7** Expected to observed β-diversity ratio ($\overline{\beta_{exp}:\beta_{obs}}$), derived from null-model analysis. Values below 1 (dashed line) indicate that the observed β-diversity is greater than expected by chance under the null model, while values above 1 indicate the opposite. Rhombuses represent 16S rRNA gene data at the bacterial ASV level. Circles represent shotgun metagenomics data at the bacterial genus (open circles) and lowest IP2G gene (closed circles) levels. Values were calculated using all taxa/genes. Each point calculation involved all replicates ($n = 4$ for phases A and L; $n = 3$ for phase H) of each phase evaluated.

with low and high organic loading. Diversity metrics are among the fundamental descriptive variables of community ecology (46), on both a local (α-diversity) and a spatiotemporal (β-diversity) scale (47). In our study, high-organic-loading disturbance led to a reduction of the nitrite oxidation function in these reactors, coinciding with a marked community differentiation from the low-organic-loading reactors in terms of β-diversity (Fig. 2B), although only a few taxa are known to perform this function. However, we also found higher taxonomic α-diversity in the high-organic-loading reactors ($^2D_{ASV}$ and $^2D_{Genus}$) (Fig. 1), which means that the relative abundances were more evenly distributed among taxa for these communities. Since the low-organic-loading reactors displayed better chemical oxygen demand (COD) removal and complete nitrification with almost no residual $NH_4^+$-N or $NO_2^-$-N, it was expected that they would harbor more diverse communities, as community evenness was suggested to be a key factor in preserving the functional stability of an ecosystem (48). We did indeed observe higher functional gene α-diversity ($^2D_{IP2G}$) (Fig. 1C) for the low-organic-loading reactors toward the end of the study, which is similar to the opposing trends of taxonomic and functional gene α-diversity previously reported after autoclave sterilization of soil microbial communities (49). Further, the dominant genera of the acclimation phase gave way to two separate clusters with different dominant genera at the end of the study (Fig. 3; Fig. S2). Bacterial successional dynamics have been described

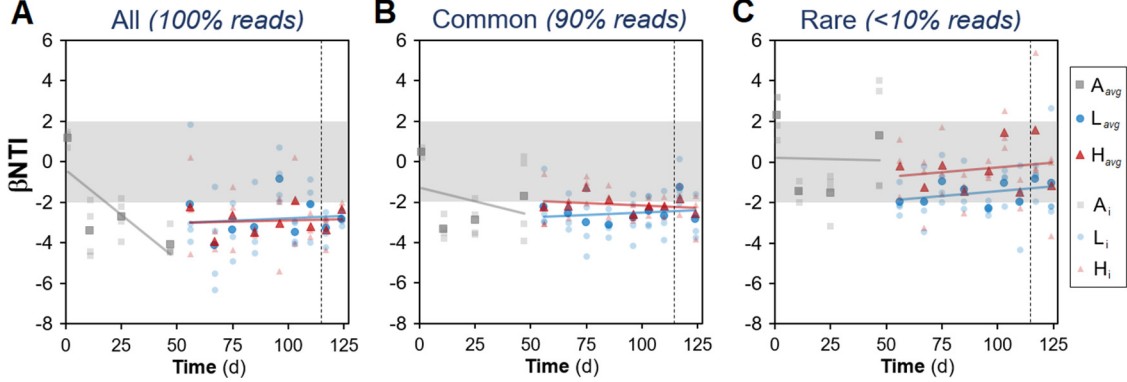

**FIG 8** Nearest taxon index (βNTI) temporal dynamics for bacterial ASVs, derived from null-model analysis. Values were calculated using all ASVs (A), common ASVs (90% accumulated reads) (B), and rare ASVs (<10% accumulated reads) (C). Phases: A, acclimation (gray squares; $n = 4$); L, low organic loading (blue circles; $n = 4$); H, high organic loading (red triangles; $n = 3$). Lines represent linear regression fitting. Zones where stochastic processes dominate ($-2 < \beta NTI < 2$) are shaded in gray. Vertical dashed lines indicate the shift from high to low organic loading. Darker symbols represent average values at a given time point.

before for activated sludge systems, but studies often present analysis at the phylum or class level of taxonomy (26, 27). Analysis of broad taxonomic categories may hide important assembly mechanisms operating at lower taxonomic levels (50), which is why we present successional changes at the genus level in this study. For example, the two main genera across low- and high-organic-loading reactors (Fig. 3) were *Thauera* and *Paracoccus*, which are denitrifying organisms in activated sludge systems (51). These genera likely benefited from their versatility in carbon substrate uptake and ability to reduce available nitrite and nitrate during the anoxic phase of the bioreactor cycle (52). The accumulation of nitrite during the aerobic phase in high-organic-loading reactors seemed to have benefited *Paracoccus* more than *Thauera*. The main nitrifying genera were *Nitrosomonas* and *Nitrospira*, whose relative abundance was diminished in high-organic-loading reactors, consistent with a reduction in the nitrite oxidation function (details in reference 45).

We further evaluated the succession of trait complexes, which are a product of the expression of multiple true traits (53), to identify patterns suggesting differential functional gene investment at different stages of the study. Traits like cell motility and cell wall were enriched in low-organic-loading reactors, while traits of replication and repair prevailed in high-organic-loading ones (Fig. 4). The enrichment of maintenance functions across disturbed reactors exemplifies how organisms have to invest resources to adapt to changes in the environment (54). High-organic-loading reactors also showed an increased prevalence of ATP-binding cassette transporter (Fig. S3A and B) and stress response (Fig. S3C) genes, which encode traits related to cell survival. The prevalence of certain functional genes suggests community-level tradeoffs under disturbance similar to the ones described by life history strategy theory (55), comparable to those reported for sludge bioreactors under pollutant disturbance (56). Taken together, these observations highlight how organisms face tradeoffs when allocating resources to certain traits to maximize their fitness, which depends on abiotic and biotic interactions within their habitat (57, 58).

Additionally, two of the disturbed reactors recovered the nitrite oxidation function after returning to low-organic-loading conditions for 14 days (Fig. 5). However, the $\alpha$- and $\beta$-diversity with respect to taxonomy and functional genes did not revert to previous levels (Fig. 1 and 2). The fact that an altered community was still able to provide the functions of the original one (carbon removal and nitrite oxidation) supports the notion of functional redundancy (59) in complex microbial systems like bioreactors, where different types of organisms are capable of performing a wide range of functions (60).

**Community assembly mechanisms differ for taxa and genes, as well as for common and rare fractions.** In terms of relative deterministic strength, disturbed reactors showed a stronger role of deterministic mechanisms for all three types of data sets evaluated (Fig. 6), likely due to the selective pressure via environmental filtering (61) imposed by the disturbance. Deterministic mechanisms dominated bacterial community assembly at the ASV taxonomic level across all reactors and were stronger under high organic loading (Fig. 6A). This agrees with previous studies of press disturbance in mesocosm sludge bioreactors using a micropollutant (3-chloroaniline), which reported higher similarity among disturbed reactors than among control reactors via 16S rRNA gene amplicon analysis (62, 63), although without quantifying assembly mechanisms. Partitioning the contribution of common (90% of the reads accumulated) and rare (<10% of the reads accumulated) portions of the data revealed that the common fraction was significantly driven by deterministic assembly mechanisms (Fig. 6B and 8B), while the rare portion displayed high stochasticity (Fig. 6C and 8C). Stochastic mechanisms of ecological drift can operate in the portion of taxa at low relative abundances (20), while the overall community assembly is mainly deterministic. These findings are in agreement with prior studies that reported higher variability for rare taxa than common taxa in full-scale and mesocosm bioreactor sludge systems (26–29), albeit without the use of null-model analysis. Although dissimilarity observations can highlight deterministic effects, they have low power for inferring stochasticity

(23), which is why we partitioned community assembly mechanisms via null-model analysis in the present work. Similarly, a recent study on granular biofilm reactors using one simple carbon source also reported stronger homogeneous selection for abundant taxa and higher stochastic assembly via drift for low-abundance taxa (38). This was done using the type of null model that was applied on the metabarcoding data set in this study (Fig. 8), and it has the advantage of incorporating phylogeny into the analysis (64) but does not take advantage of replicated designs the way the null model of Kraft et al. (65) does. As the results from null-model analyses are very sensitive to the models, algorithms, and diversity metrics employed (66), concordant outcomes in studies using different approaches are desired, and this requires more research (23). Further, the aforementioned studies employed 16S rRNA gene metabarcoding with OTU-type clustering for their community analyses. Here, we inferred community assembly mechanisms from ASV data, which has several benefits over traditional OTU clustering (67), including generating 10 to 100 times fewer spurious units (68), which reduces bias for $\alpha$-diversity estimations. Although classification criteria for rare and common fractions are arbitrary (30, 69), there is consistency between our community assembly observations from 16S rRNA gene metabarcoding data and the current literature. Nonetheless, further research is needed to evaluate whether the rare fraction is biologically active (70).

Metagenomics data also revealed higher deterministic strength at high organic loading for taxa (Fig. 6D) and slightly higher DS for genes (Fig. 6G). This concurs with the higher deterministic assembly reported previously for sludge microcosm reactors that were press disturbed with 3-chloroaniline and compared to undisturbed ones, via null-model analysis on metagenomics genus-level data (21). The study included a different type of press disturbance in a bioreactor scale 2 orders of magnitude smaller (20 ml) than the ones in this study (5 liters), and also in a shorter time span (35 days). Since diversity is multidimensional and scale dependent (71), assembly mechanisms and the results of null-model analyses can differ at different spatial and temporal scales (20, 66); therefore, these concurring effects of press disturbance in community assembly that were observed at microcosm and mesocosm bioreactor scales are relevant. Indeed, it is necessary to quantify the relative importance of stochastic and deterministic assembly mechanisms for microbial communities across different spatial and temporal scales, environmental gradients, systems, and types of disturbance (20, 23).

Most of the DS points for the overall IP2G gene data set fell below 50% (Fig. 6G), implying that stochastic mechanisms of assembly were stronger for genes than for taxa. Both data sets used the same number of individuals per sample for the null-model analysis (100,000) and thus represent a balanced sampling effort for such a comparison. Separate evaluation of the common and rare fractions in terms of DS showed that deterministic assembly mechanisms were stronger in the common fraction (Fig. 6E and H), while the rare portion was more influenced by stochasticity (Fig. 6F and I), which is similar to what we observed for ASV data using two different null-modeling approaches. Further, $\beta$-diversity was always greater than expected for bacterial taxa under the null model regardless of the sequencing method (Fig. 7; Table S3), suggesting that taxa tended to be more aggregated within replicate bioreactors than expected by chance. Aggregation can be explained by processes of habitat filtering (72) due to the recurrent conditions at each low and high organic loading level, as well as by dispersal limitation (73), which is a condition of the experimental design using reactors as closed systems without immigration. However, the opposite was shown for the functional gene data set, where the mean expected $\beta$-diversity of individual IP2G genes under the null model was higher than the observed $\beta$-diversity; i.e., replicate bioreactors were more similar than expected by chance (Fig. 7). These results highlight that functional gene data can indicate patterns of dominant stochastic versus deterministic mechanisms of community assembly that differ from patterns obtained after analysis of taxon data.

**Sequencing methods, diversity, and assembly mechanisms: what is consistent and what is not.** Using two different community profiling approaches on the same samples, we found that press disturbance favored deterministic assembly mechanisms, where bioreactor bacterial communities at low and high organic loading levels clearly

separated into two different clusters in terms of $\beta$-diversity for taxa and genes. Both methods identified a greater influence of deterministic assembly mechanisms in the common fraction of the community, whereas stochasticity was more important in the low-abundance fraction. These complementary analyses suggest that one should be able to assess the effect of a press disturbance on both $\beta$-diversity and the overall assembly mechanisms of microbial communities, regardless of the sequencing approach employed and whether the focus is on taxa or functional genes.

However, methodologies were inconsistent when the relative contributions of deterministic and stochastic mechanisms of community assembly were probed, with metagenomics data displaying higher stochasticity for genera than when ASVs from 16S rRNA gene metabarcoding were used. This was not an effect of assessing different levels of taxonomic resolution, as higher resolution levels were shown to be more conserved than lower ones (50) and therefore displayed assembly mechanisms that were more deterministic. However, this could have been the effect of a different sampling coverage under the null-model analysis, because metabarcoding renders a lower total number of reads per sample, about 1 order of magnitude, than metagenomics analysis. When the focus was only on metagenomics data, functional gene assembly was found to be more stochastically driven than that of taxa at the genus level, even under a balanced sampling coverage. Further, there were differences in terms of which fraction of the community, common or rare, affected overall community assembly. Overall functional gene assembly appeared to be driven by a balance between common and rare fractions. On the other hand, overall assembly of bacterial taxa had DS values similar to those of the rare fraction and $\beta$NTI values similar to those of the common fraction. Similarly, the effect of disturbance on $\alpha$-diversity was different for genes and taxonomic data. The challenge of reconciling results from different sequencing methods has been recognized and requires further research (41). Differences in assembly mechanisms can arise due to the distinct biological signature being evaluated when a fraction of a specific DNA marker like the 16S rRNA gene is targeted, compared to using DNA of the whole meta-community as the basis of classification of taxa and functional genes. Currently, the 16S rRNA amplicon approach is expected to result in less ambiguity, because fewer singletons are generated than with shotgun metagenomics. Not surprisingly, the majority of existing studies on microbial community assembly focused on taxonomic OTU data sets from 16S rRNA gene metabarcoding (17, 18, 33–38), with very few studies also using shotgun metagenomics data to quantify community assembly mechanisms in bioreactor systems for wastewater treatment (21). Future studies on the impact of disturbance in assembly mechanisms would benefit from also incorporating whole-community DNA information, enabling the assessment of less conserved genes that could unveil important aspects of community assembly (40). The latter is promising given the increasing availability of methods that are faster, are PCR independent, and allow long-read sequencing (74).

**Concluding remarks.** The joint evaluation of assembly mechanisms, community structure, and function of bacterial taxa and functional genes provided in-depth understanding of the response of complex microbial systems to a doubling of the organic loading rate. This press disturbance altered community function, structure, and assembly mechanisms. Disturbance had an effect not only on community function but also on its functional potential, emphasizing the relevance of assessing communities of organisms together with communities of genes. Through null-model analyses, community assembly was found to have a stronger deterministic component in the common fraction, whereas the role of stochastic mechanisms was higher for the less abundant portion of the community. Also, reactors that recovered functions after returning to low-organic-loading conditions maintained different $\alpha$- and $\beta$-diversities compared to reactors that had not been disturbed, in terms of both taxonomic and functional genes, showing that resilience based on community function does not necessarily translate into resilience based on community structure. Further, we urge caution when assessing microbial community assembly mechanisms, as results can vary

depending on the approach (16S rRNA gene metabarcoding and shotgun metagenomics taxonomic or functional gene community profiling) and whether the focus is on taxa or genes. Not only can genes suggest different dominance patterns of stochastic versus deterministic mechanisms of community assembly compared to taxa, but the fraction of the community driving such assembly mechanisms—common versus rare—can also differ. Finally, this study employed alteration in the substrate feeding scheme as a type of press disturbance. More research is needed on different types of disturbances (e.g., pollutant additions, pH shifts, and temperature changes) within different complex microbial systems at different scales to broadly validate our observations. Additional studies covering different spatial and temporal scales, environmental gradients and types of disturbance could lead to a general framework of how press disturbances alter the structure, function, and assembly mechanisms of microbial communities.

## MATERIALS AND METHODS

**Experimental design.** Sequencing batch reactors (SBR) with a 5-liter working volume were inoculated with activated sludge from a WWTP in Singapore and operated on continuous 12-h cycles with intermittent aeration. The experimental setup was detailed elsewhere (45). Initially, four reactors were acclimated to laboratory conditions for 53 days while being fed a complex synthetic wastewater containing a variety of carbon and nitrogen compounds. On day 54, the biomass of the acclimation reactors was thoroughly mixed and redistributed across eight reactors. From these, four were randomly selected and designated high-organic-loading reactors, receiving double the carbon substrate in terms of chemical oxygen demand (COD) in the feed (629 mg COD liter$^{-1}$ and 100 mg total Kjeldahl nitrogen [TKN] liter$^{-1}$) as a press disturbance for 60 days. One of these reactors broke prematurely, reducing the count to three replicates. The remaining four reactors were operated under low-organic-loading conditions (323 mg COD liter$^{-1}$ and 92 mg TKN liter$^{-1}$). During the last 2 weeks of the study (days 114 to 127), the feed for the high-organic-loading reactors was reduced to equal that of low-organic-loading reactors (details in Table S1). SBR stages were 5 min of feeding, 200 min of anoxic/anaerobic reaction, 445 min of aerobic reaction, 50 min of sludge settling, and 20 min of supernatant draining in each cycle. Temperature was controlled at 30°C, pH at 6 to 9, and dissolved oxygen concentration at 2 to 6 mg liter$^{-1}$ (during the aerobic phase). Two cycles per day accounted for a hydraulic retention time of 24 h. Effluent and influent compositions were measured 2 or 3 times per week in accordance with standard methods (75). The targets were soluble COD, total alkalinity, and nitrogen species (ammonium, nitrite, and nitrate), as well as TKN in the liquid phase, using colorimetric tests and ion chromatography. Sludge biomass was measured as total suspended solids (TSS) twice a week, after which biomass removal was done to target 1,500 mg liter$^{-1}$ of TSS in order to control the food-to-biomass ratio (F:M) (Table S1). See reference 45 for detailed information on the source WWTP, inoculum collection, feed preparation, sludge sampling, analytical methods, DNA extractions, and equations for calculation of operational parameters.

**16S rRNA gene metabarcoding and read processing.** We used primer set 341f/785r, which targets the V3-V4 variable regions of the bacterial 16S rRNA gene (76). The libraries were sequenced in house on an Illumina MiSeq system (v.3) with 20% PhiX spike-in, at a 300-bp paired-end read length. Sequenced sample libraries were processed with the dada2 (v.1.3.3) R package (68), which allows inference of ASVs (67). Illumina adaptors and PCR primers were trimmed prior to quality filtering. Sequences were truncated after 280 and 255 nucleotides for forward and reverse reads, respectively. After truncation, reads with expected error rates higher than 3 and 5 for forward and reverse reads, respectively, were removed. Reads were merged with a minimum overlap of 20 bp. Chimeric sequences (0.18% on average) were identified and removed. For a total of 79 samples, an average of 19,331 reads were kept per sample after processing, representing 49.3% of the average input reads. Taxonomy was assigned using the SILVA database (v.132) (77). See reference 45 for further details. Samples were rarefed to the lowest number of reads (7,704) in a sample after processing (Fig. S6A and B).

**Metagenomics sequencing and read processing.** Libraries were sequenced in house on an Illumina HiSeq2500 system in rapid mode at a 250-bp paired-end read length. In total, around 292 million paired-end reads were generated, with 3.4 ± 0.4 million paired-end reads on average per sample (total of 43 samples). Illumina adaptors, short reads, low-quality reads, and reads containing any ambiguous bases were removed using cutadapt (78). High-quality reads (91.0% ± 1.4% of the raw reads) were randomly subsampled to an even depth of 4,678,535 for each sample prior to further analysis. Taxonomic assignment of metagenomics reads was done as previously described (56). High-quality reads were aligned against the NCBI nonredundant (NR) protein database (March 2016) using DIAMOND v.0.7.10.59 (79). The lowest-common-ancestor approach implemented in MEGAN Community Edition v.6.5.5 (80) was used to assign taxonomy to the NCBI-NR-aligned reads. On average, 36.8% of the high-quality reads were assigned to cellular organisms, of which 98.4% were assigned to the bacterial domain. Functional potential data were also obtained from the metagenomics data set using MEGAN. We employed the four databases available, with the rate of assigned/total reads being the highest for the IP2G database (52%), followed by SEED (21%), COG (15%), and KEGG (13%). See reference 45 for more details. Samples were rarefed to the lowest number of genus-level summarized reads (537,616) in a sample after processing (Fig. S6C and D).

**Bacterial community analysis and statistics.** All reported $P$ values for statistical tests in this study were corrected for multiple comparisons using a false discovery rate (FDR) of 5% (81). Community

mSystems®

structure was assessed by a combination of nonmetric multidimensional scaling (NMDS) ordination and multivariate tests of permutational analysis of variance (PERMANOVA) and permutational analysis of dispersion (PERMDISP) on Bray-Curtis dissimilarity matrixes constructed from square-root-transformed normalized abundance data using PRIMER (v.7) (82). Hill diversity indices (83) were employed to quantify $\alpha$-diversity as described elsewhere (21, 84). Local polynomial regression fitting was applied using the loess function from the ggplot2 package in R (85), including 95% confidence intervals. Welch's ANOVA was used for univariate testing. Bacterial genera, functional genes from the IP2G database at the lowest gene assigned level (metagenomics data set), and ASVs (16S rRNA gene amplicon data set) were employed for calculating $\alpha$- and $\beta$-diversity metrics and for quantifying assembly mechanisms.

**Null-model analyses on diversity.** The effect of underlying assembly mechanisms was assessed through a mathematical null model which assumes that species interactions are not important for community assembly (32) and quantifies the effective bacterial turnover expressed as a proportion of total bacterial diversity. It was developed for woody plants (65) and recently applied to sludge (21) and groundwater (43) microbial communities. The model defines $\beta$-diversity as the $\beta$-partition ($\beta = 1 - \overline{\alpha}/\gamma$). To adapt it to handle microbial community data, we considered "species" in the model as ASVs, genera, and genes at the lowest IP2G gene level, while each individual count was one read within the corresponding data set. The model randomizes the location of each individual within the independent replicate reactors for each of the low- and high-organic-loading levels while maintaining the total quantity of individuals per reactor, the relative abundance of each species (i.e., ASV, genus, or IP2G gene), and the $\gamma$-diversity. This way, it takes into account both composition and relative abundances. We applied the model across different time points of the experiment. Samples from metagenomics data sets for both genera and IP2G genes were normalized to 100,000 reads (equivalent to 100,000 model individuals) (21). In this manner, the number of total individuals to be shuffled in each model iteration was reduced, and the sampling effort across these data sets was equalized for null-model comparisons across metagenomics data. Samples from the 16S rRNA gene ASV data set were analyzed at their rarefied values of 7,704 reads.

Each step of the null model calculates expected mean $\alpha$-diversities per treatment level and then estimates an expected $\beta$-partition. After 10,000 repetitions, the means of the distribution of random $\beta$-partitions ($\overline{\beta_{exp}}$) for each treatment level are calculated. The relative contribution of deterministic assembly mechanisms is then quantified using the deterministic strength (DS) metric, which measures the deviation of the observed $\beta$-diversity compared to that expected by chance. DS is equal to the absolute value of the difference between the observed ($\beta_{obs}$) and mean expected $\beta$-diversity, divided by the observed $\beta$-diversity: $DS = |\beta_{obs} - \overline{\beta_{exp}}|/\beta_{obs}$. It is the complement of the stochastic intensity (SI) metric ($SI = 1 - |\beta_{obs} - \overline{\beta_{exp}}|/\beta_{obs}$) defined previously (21). Higher values of DS indicate a higher deviation of the observed $\beta$-diversity from the null $\beta$-diversity expectation, thus suggesting a stronger effect of deterministic-based mechanisms. Contrarily, lower DS values indicate a smaller difference between observed and null $\beta$-diversities, suggesting a more important role of stochastic mechanisms of assembly. Additionally, the expected-to-observed $\beta$-diversity ratio ($\overline{\beta_{exp}}:\beta_{obs}$) was calculated; values below 1 indicate that the observed $\beta$-diversity is greater than expected by chance under the null model, while values above 1 indicate the opposite.

To further assess community assembly of common and rare fractions, all three data sets were arbitrarily partitioned. For each sample, taxa or genes falling into the 90% abundance rank of accumulated reads of the data set were classified as common, while the remaining 10% were classified as rare. Common and rare fractions from all samples were then compiled to generate two separate matrices. This was a conservative approach to offset bias when employing relative abundances to quantify taxa or genes. To ensure adequate community coverage in terms of a balanced sampling effort across both fractions prior to null-model analysis, the counts per sample of the common matrix were normalized to equal that of the rare matrix. This accounted for potential biases in DS estimations from the null model, where taxa with many "individuals" (reads) tend to be distributed in all "plots" (bioreactors) when the number of the latter is small compared to the number of individuals. Such normalization was done in all three data sets without losing any community member (i.e., the total number of ASVs/taxa/genes within the common fraction remained constant), while also leaving relative abundances unaltered.

We further assessed assembly mechanisms by using an alternative phylogenetic-based null-modeling approach on the metabarcoding ASV data set. The model uses the $\beta$-mean nearest taxon distance ($\beta$MNTD) (86), which quantifies the phylogenetic distance between ASVs in one community, as a measure of the clustering of closely related ASVs. Phylogenetic relatedness of ASVs was characterized by multiple alignment of ASV sequences using the decipher R package (87) (v.2.14.0). The phylogenetic tree was then constructed, and a GTR+G+I (generalized time reversible with gamma rate variation) maximum likelihood tree was then fitted using the phangorn R package (88) (v.2.5.5). To quantify the degree to which $\beta$MNTD deviates from a null-model expectation, ASVs and abundances were shuffled across the tips of the phylogenetic tree. After shuffling, $\beta$MNTD was recalculated to obtain a null value, and repeating the shuffling 1,000 times provided a null distribution. The difference between observed $\beta$MNTD and the mean of the null distribution was measured in units of standard deviation, which is referred to as the $\beta$-nearest taxon index ($\beta$NTI) (64). A $|\beta NTI|$ of >2 indicates that the observed turnover between a pair of communities is significantly deterministic, while a $|\beta NTI|$ of <2 suggests stochastic assembly (18). This analysis was done using the phylocom R package (89). To support the assumption of significant phylogenetic signal, phylogenetic Mantel correlograms were constructed relating between-ASV niche differences to between-ASV phylogenetic distances across a given phylogenetic distance, following the method previously described (18, 64). Environmental niches were constructed from effluent data (soluble

chemical oxygen demand; ammonium, nitrite, and nitrate as nitrogen). Phylogenetic distances were quantified for 50 phylogenetic distance bins, and significance of Pearson correlations was evaluated using 1,000 permutations and FDR (5%) correction.

**Data availability.** DNA sequencing data are available at NCBI BioProject under accession no. PRJNA559245. See the supplemental material for details about community structure dynamics based on bacterial genera and functional genes. R script to replicate the null-model analysis and all other relevant data can be publicly accessed on Figshare (https://doi.org/10.6084/m9.figshare.12326330).

## SUPPLEMENTAL MATERIAL

Supplemental material is available online only.

**TEXT S1**, PDF file, 0.03 MB.
**FIG S1**, TIF file, 0.8 MB.
**FIG S2**, PDF file, 0.3 MB.
**FIG S3**, PDF file, 0.1 MB.
**FIG S4**, TIF file, 0.6 MB.
**FIG S5**, TIF file, 0.7 MB.
**FIG S6**, TIF file, 0.3 MB.
**TABLE S1**, PDF file, 0.2 MB.
**TABLE S2**, PDF file, 0.3 MB.
**TABLE S3**, PDF file, 0.3 MB.

## ACKNOWLEDGMENTS

This research was supported by the Singapore National Research Foundation and Ministry of Education under the Research Centre of Excellence Program. E.S. was partially supported by a Fulbright Fellowship.

We thank D. I. Drautz-Moses for her support with the 16S rRNA gene amplicon and metagenomics library preparation and sequencing pipelines employed. W. X. Phua is acknowledged for her support with bioreactor operation. We thank T. J. Qiang and N. A. B. A. Latiff for their assistance with molecular work. We thank the three anonymous reviewers for their comments and suggestions.

E.S. and S.W. conceived the study. E.S. designed the experiment, and S.W. obtained the funding for the study. E.S. performed the experiments and the 16S rRNA gene bioinformatics analyses. F.C. did the metagenomics and contributed to the metabarcoding bioinformatics analyses. E.S. and F.C. performed the null-model analyses. E.S. interpreted the data and elaborated the main arguments in the manuscript. E.S. and S.W. wrote the manuscript.

We declare no competing interests.

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
