## [Reviewer comments · mSystems]

Press disturbance alters community structure and assembly mechanisms of bacterial taxa and functional genes in mesocosm-scale bioreactors

Ezequiel Santillan, Florentin Constancias, and Stefan Wuertz

Corresponding Author(s): Stefan Wuertz, University of California, Davis

Review Timeline:

Submission Date:	May 29, 2020
Editorial Decision:	June 24, 2020
Revision Received:	August 1, 2020
Accepted:	August 3, 2020

Editor: Sean Gibbons

Reviewer(s): Disclosure of reviewer identity is with reference to reviewer comments included in decision letter(s). The following individuals involved in review of your submission have agreed to reveal their identity: Jake Valenzuela (Reviewer #1)

Transaction Report:

DOI: <https://doi.org/10.1128/mSystems.00471-20>

Revised submission of mSystems00214-20

Hereby we address point-by-point the comments from the three referees in the exact order they appear in the original referee report (included at the end of this document).

Reviewer #1:

Rev. 1 - Comment 1: Please define and discuss the ecological theories that are referenced. There are many sub-fields to microbial ecology and overlapping focus areas that are not familiar with some of the theories. This is a great opportunity to discuss how your work compares or contrasts with what has been postulated theoretically so don't assume all readers are up-to-date.

Response: We are thankful to the reviewer for his/her advice and hope that the revised version of the manuscript has reached the desired level of detail in the Discussion section. Detailed responses to his/her individual concerns are given below.

Rev. 1 - Comment 2: The manuscript by Santillan *et al.* expands on the notion that understanding the mechanisms controlling community diversity, functions, succession, is poorly understood in microbial ecology. In particular, the authors set out to test the effect of press disturbances on community assembly mechanisms through the use of large scale (5 L) batch reactors supporting full-scale treatment plant sludge over 124 days. The press disturbance was an increase in carbon load (H) compared to the control level (L). The researchers used shotgun metagenomics, 16S rRNA gene metabarcoding, effluent chemical characterization, and a null model approach to conclude that deterministic assembly was a stronger driver in the disturbed reactors (H) due to the common taxa. Importantly, the use of functional gene data compared to taxon data resulted in different community assembly patterns. Overall the paper reads well and has done the proper statistical analysis and provides thorough supplemental information.

Response: We thank the reviewer for this comment.

Rev. 1 - Comment 3: The major issues with the manuscript are not in any flawed technique or conclusions, however, there is a lack of discussion around the theories in which they are testing leaving the reader wanting in-depth characterizations of the mechanistic underpinnings from their conclusions.

Response: We thank the reviewer for his/her suggestion to expand the discussion around the theories tested in the revised version of the manuscript. Detailed responses to his/her individual concerns are given below.

Rev. 1 - Comment 4: For instance, the authors should speculate on actual biology. Why are certain organisms driving the observed changes? What specific genes are accounting for the different community patterns observed? While this may not have been their major objective, finding reasoning behind the biological observations can inform the results of their statistical analyses.

Response: We agree that providing examples of biological functions or processes would help clarify the results of the statistical analyses done in this study. Following the reviewer's suggestion with regards to functional genes (which is also related to "Rev. 3 - Comment 5" below), the revised version of the manuscript includes a heat map showing successional

clusters of the 19 most abundant functional gene categories from the COG database (>10,000 reads) in reactors over time. This is now shown in Fig. 4 in lines 680-687 (page 27) of the manuscript:

Fig. 4. Successional clusters of the 19 most abundant functional gene categories from the COG database (>10,000 reads) across reactors over time. Z-scores denote how many standard deviations of the mean each sample contains (assigned reads per gene category, across all samples). Column legend represents day number and replicate reactors. Phases: A, acclimation ($n = 4$); L, low organic loading ($n = 4$); H, high organic loading ($n = 3$). Rectangles highlight groups of functional gene categories prevailing at different phases. Dashed vertical lines separate time points within the same phase. Dashed horizontal lines separate the four biggest clusters of trait complexes.

The discussion about changes in key functional genes across reactors was also expanded (which is also supported by supplementary Figures S3A-C) as follows:

Lines 192-203 (page 9): “We further evaluated the succession of trait-complexes, which are a product of the expression of multiple true traits⁵³, to identify patterns suggesting differential functional gene investment at different stages of the study. Traits like cell motility and cell wall were enriched in low organic loading reactors, while traits of replication and repair prevailed in high organic loading ones (Fig. 4). The enrichment of maintenance functions across disturbed reactors exemplifies how organisms have to invest resources to adapt to changes in the environment⁵⁴. High organic loading reactors also showed an increased prevalence of ATP-binding cassette transporter (Figs. S3A-B) and stress response (Fig. S3C) genes, which encode traits related to cell survival. The prevalence of certain functional genes suggests community-level tradeoffs under disturbance similar to the ones described by life-history strategy theory⁵⁵, comparable to those reported for sludge bioreactors under pollutant disturbance⁵⁶. Taken together, these observations highlight how organisms face tradeoffs when allocating resources to certain traits to maximize their fitness, which depend on abiotic and biotic interactions within their habitat^{57,58}.”

Following the reviewer’s suggestion with regards to specific microorganisms (which is also related to “Rev. 3 - Comment 4” below), the revised version of the manuscript now includes a discussion about the role of *Paracoccus* and *Thauera* in this ecosystem and a reference to dynamics in nitrifiers.

Lines 184-191 (page 9): “For example, the two main genera across low and high organic loading reactors (Fig. 3) were *Thauera* and *Paracoccus*, which are denitrifying organisms in activated sludge systems⁵¹. These genera likely benefited from their versatility in carbon

*substrate uptake and ability to reduce nitrite and nitrate available during the anoxic phase of the bioreactor cycle*⁵². The accumulation of nitrite during the aerobic phase in high organic loading reactors seemed to have benefited *Paracoccus* more than *Thauera*. The main nitrifying genera were *Nitrosomonas* and *Nitrospira*, whose relative abundance was diminished in high organic loading reactors, coherent with a reduction in the nitrite oxidation function (details in Santillan *et al.*⁴⁵)." To avoid overextending the discussion section by moving too much away from the community-level focus desired for this study, we have chosen to keep the section *Bacterial genus-level structure dynamics* as supplementary text in lines 23-32 (page 2 of SI).

Rev. 1 - Comment 5: Based on my review I would recommend publication if major revisions are made to explain more of the ecological theories and analysis governing their conclusions as well as increased biological interpretation.

Response: We appreciate the advice, hoping that the revised version of the manuscript has now reached the desired level of detail in within the discussion section. Detailed responses are given below.

Main critique

Rev. 1 – Major Point 1: If the authors expected that press disturbances would affect the community function, wouldn't an assessment of the actual gene expression (metatranscriptomics) provide an important dataset that can help evaluate community function? Wouldn't pairing metagenomics and metatranscriptomics combine to provide biologically meaningful insights on microbiome function? Please address.

Response: We thank the reviewer for this question. We did not check for gene expression because it was beyond the scope of our study. However, we checked for differences in genes present across different disturbance levels over time (also referred to throughout the text as functional genes or functional potential). We state the relevance of this approach in lines 88-91 (page 5): "*Microbial communities can also be assessed in terms of their functional gene structure*⁴², and while functional gene assembly can be assessed via microarray^{22,43} or clone library⁴⁴ approaches, a more complete view of the functional gene structure of the microbial community can be obtained through shotgun metagenomics."

In this study, functional genes are assessed in terms of patterns of α - (Fig. 1C) and β -diversity (Fig. 1D) and assembly mechanisms (Figs. 5G,H,I and 6). Further, following the reviewer's suggestion, functional gene dynamics are now shown in Fig. 4 and Fig. S3A-C, including a discussion in lines 192-203 (page 9). We focused on trait complexes which are a product of the expression of multiple true traits (see Crowther, T. W. *et al.* 2014. Untangling the fungal niche: the trait-based approach. *Front. Microbiol.* 5, 1-12), grouping different sets of genes into categories. Such an analysis is important because organisms face tradeoffs when allocating resources to certain traits to maximize their fitness, which depend on abiotic and biotic interactions within their habitat (see Santillan *et al.* 2019. Trait-based life-history strategies explain succession scenario for complex bacterial communities under varying disturbance, *Env. Micro.* 21(10), 3751-3756).

Therefore, we believe this study clearly shows a differentiation in functional potential across different levels of disturbance, providing biologically meaningful insights on microbiome function, without having to evaluate gene expression. Finally, the way functional potential is presented and discussed has also been improved in the revised version, following the

reviewer’s suggestions (for details see *Response to Rev. 1 - Comment 4* and *Response to Rev. 3 – Comment 5* in this letter).

Rev. 1 - Major Point 2: Could the authors discuss the potential role of the press disturbance as a selection pressure that could select for variant strains of the community to emerge? Was there any evidence for strain selection over time?

Response: This is an inspiring idea. Yet, variant strain selection was beyond the scope of the manuscript, which is focused on structure, function, and assembly of bacterial communities. The main focus of analysis and discussion is aimed at community-level patterns. Following the reviewer’s suggestions, the revised version includes a discussion about changes of specific functional gene categories, supported by Fig. 4 and Fig. S3, and main genera, supported by Fig. 3 and Fig. S2 (see “*Response to Rev. 1 – Comment 4*”, “*Response to Rev. 3 - Comment 4*” and “*Response to Rev. 3 - Comment 5*”).

In addition, the potential role of the press disturbance as a selection pressure that could select for variant strains of the community to emerge is discussed for the case of nitrifiers in the related reference 45 (Santillan *et al.* DOI: 10.1101/605733v4). The changes in abundance of different *Nitrosomonas* ASVs suggested a succession of organisms within this genus. Additionally, through the use of exact sequence variants we observed that *Nitrosomonas* was the most diverse nitrifying genus with 21 different ASVs being detected. In this context, different ASVs are an indication of potentially different strains, although to have strain-level resolution further analysis has to be done. The figure shown below is included in Ref. 45 where this aspect is shown; please note how *Nitrosomonas europaea* (left central panel) is selected within press disturbed reactors from d75 onwards.

Fig. S2. Temporal relative abundance of ASVs assigned to nitrifier genera in each reactor. Phases: A, acclimation (n = 4); L, low F:M:C:N (n = 4); H, high F:M:C:N (n = 3). Vertical dashed line indicates the shift from high to low F:M:C:N. The abundance rank number (among all 1646 ASVs detected) is shown in parentheses. Brackets indicate the number of other additional lower abundance ASVs summed as ‘others’ within a panel. The higher number of ASVs detected suggests that AOB populations were more diverse than NOB populations.

Hence, we have addressed the suggestion made by the reviewer in the new submission as explained above, while also referring to Ref 45 for further examples.

Rev. 1 - Major Point 3: Please add more background to why α - and β -diversity are important assessment tools?

Response: Following the reviewer's wishes, we have added more background on why α - and β -diversity are important assessment tools as follows:

Lines 165-167 (page 8): “Community function and structure, in terms of α - and β -diversity, were clearly different in low and high organic loading reactors. Diversity metrics are among the fundamental descriptive variables of community ecology⁴⁶, at both a local (α -diversity) and spatiotemporal (β -diversity) scale⁴⁷.”

Rev. 1 - Major Point 4: Please provide more reasoning on why you chose to display the 2nd order true α - diversity, and not the zero or first order α -diversities like in S1.

Response: We thank the reviewer for this observation. Indeed, in Figure 1 we chose to display only the 2nd order true α -diversity (2D) because of two reasons: (i) For complex communities there is often a huge difference between the abundance of rare and abundant taxa, and overall changes in communities are driven by changes in the most abundant taxa (see *van Dorst, J. et al. 2013. Community fingerprinting in a sequencing world. FEMS Microbiol. Ecol. (2014) 89, 316-330*), which in terms of α -diversity are best described by 2D (see *Haegeman, B. et al. Robust estimation of microbial diversity in theory and in practice. ISME J. 7, 1092-1101*). (ii) The use of richness (first order α -diversity, 0D) to describe microbial communities is not reliable since it is heavily constrained by the method of measurement (again see *Haegeman, B. et al. 2013*), which makes the comparison of results from different sequencing techniques using this metric meaningless (see *Shade, A. 2017. Diversity is the question, not the answer. ISME J. 11, 1-6*). However, we also acknowledge that the scientific community working in microbial ecology is accustomed to seeing richness metrics, the reason why we also provide that information as supplementary material. To follow the reviewer's suggestion of providing more reasoning for the reader on why we made such a choice, we have now included a subsection in the supplementary text as follows:

Lines 33-45 (pages 2-3 of SI): “Richness is not recommended as α -diversity metric for microbial community studies. In Fig. 1 we chose to display only the 2nd order true α -diversity (2D) because of two reasons: (i) For complex communities there is often a huge difference between the abundance of rare and abundant taxa⁴, which in terms of α -diversity are best described by 2D α -diversity metric⁵. (ii) The use of richness (first order α -diversity, 0D) to describe microbial communities is not reliable since it is heavily constrained by the method of measurement⁵, which makes the comparison of results from different sequencing techniques using this metric meaningless⁶. For example, the increase in ${}^0D_{Genus}$ diversity for low organic loading reactors during the disturbance phase (Fig. S1D) likely means that more genera increased in abundance above the limit of detection, as new organisms could not be incorporated since our bioreactors system was closed to immigration. The latter implies that these taxa were already in the reactors but at levels below the detection limit. This phenomenon could happen during any study on complex microbial communities leading to spurious patterns without ecological meaning. It is, therefore, not recommended to draw conclusions based on bacterial richness dynamics.”

Rev. 1 - Major Point 5: The correlation Pearson's correlation vectors of the nitrite and nitrate in Fig. 2B appears with no supporting information in the main text. Please divulge on its meaning in the Results section and why it was added to the panel.

Response: We thank the reviewer for noticing; this was omitted by mistake in our initial submission. In the Results section, this should have been mentioned within the subsection "Dynamics of ecosystem functions". The revised version of the manuscript has now this corrected as follows:

Lines 135-137 (page 7) "...there was a clear distinction between reactors receiving low and high organic loading, with the disturbed reactors displaying partial nitrification with high NO_2^- -N concentrations in the effluent (Fig. S4, Fig. 2B)."

These vectors were added in Fig. 2B to help the reader identify at a glance which group of reactors had high nitrite or high nitrate in the effluent. This way, Fig. S4 would be checked only by those readers who are particularly interested in temporal effluent concentrations at the end of a reactor cycle.

Rev. 1 - Major Point 6: In Fig. 3, how were the heatmaps clustered? Burkholderiaceae-NA is not bolded but is in each phase.

Response: For each phase (A, acclimation; L, low organic loading; H, high organic loading) the top-15 genera are ordered from most abundant (top) to least (bottom) across all time points considered within the phase. We have also corrected the bolding for *Burkholderiaceae-NA*.

Rev. 1 - Major Point 7: In Fig. 5H, there are very high deterministic strengths (275%) of the common reads, yet there is no discussion on this anomaly or outlier? What does that mean, how can you have 275%?

Response: We thank the reviewer for pointing this out. The anomaly has now been fixed (Fig. 6H in the revised manuscript) due to the improved treatment of the input matrixes used for the null model analysis on the common fraction of the datasets. This was done to equalize sampling effort following the suggestions of Reviewer #2.

Extract from Lines 696-704 (page 29): "**Fig. 6.** Deterministic strength (DS) temporal dynamics for bacterial taxa and functional genes, derived from null model analysis. [...] (G-I) Metagenomics sequencing IP2G lowest gene level. [...]."

Rev. 1 – Major Point 8: Expand on the “insurance hypothesis and functional redundancy”. Discuss please, its in the discussion section.

Response: We thank the reviewer for this suggestion. The modified version of the manuscript explains the functional redundancy concept:

Lines 204-209 (page 9): *“Additionally, two of the disturbed reactors recovered the nitrite oxidation function after returning to low organic loading conditions for 14 days (Fig. 5). However, the α - and β -diversity with respect to either taxonomy or functional genes did not revert to previous levels (Figs. 1 and 2). The fact that an altered community was still able to provide the functions of the original one (carbon removal and nitrite oxidation) supports the notion of functional redundancy⁵⁹ in complex microbial systems like bioreactors, where different types of organisms are capable of performing a wide range of functions⁶⁰.”*

We have removed the ‘insurance hypothesis’ concept from the revised version as it is not needed to make our point.

Rev. 1 – Minor Point 1: Please expand on why this particular disturbance was chosen. Why not temperature, pH, antibiotic, etc.?

Response: We are thankful for this suggestion. This type of disturbance (continuous alteration in the substrate feeding scheme) was chosen because it can occur in practice when using sludge bioreactors. This is mentioned in lines 58-65 (page 4) of the introduction: *“Press disturbances that impose a long-term continuous change of species abundances by altering the environment⁸ are of interest in microbial ecology as they can drive systems to alternative stable states with different community function and structure⁹. These disturbances could occur in the form of environment modifications that are not directly harmful to organisms, while still providing less abundant community members opportunities to grow¹⁰. In sludge bioreactors, a continuous alteration in the substrate feeding scheme can trigger changes in community function and structure, yet whether these changes are reproducible¹¹ and whether they can be reversed when the disturbance ceases remains unknown”*. With regards to mentioning other types of disturbance, following the reviewer’s suggestion the revised version of the manuscript now addresses this aspect in the discussion section:

Lines 326-332 (page 14): *“Finally, this study employed alteration in the substrate feeding scheme as a type of press disturbance. More research is needed on different types of disturbances (e.g., pollutant additions, pH shifts, temperature changes) within different complex microbial systems at different scales to broadly validate our observations. Additional studies covering different spatial and temporal scales, environmental gradients and types of disturbance could lead to a general framework of how press disturbances alter the structure, function and assembly mechanisms of microbial communities.”*

Rev. 1 – Minor Point 2: What determined the length of time for acclimation, or the high and low disturbances? Why only recovery for 14 days? If extended, there could have been more of an understanding of the resilience or recovery dynamics of the community. Basically left with 1 timepoint for the metagenomics.

Response: The length of the acclimation phase (53 d) was enough to allow for important functions like nitrification and organic carbon removal to stabilize across reactors as suggested in the literature (see Falk and Wuertz, 2010. *Water Res.* 44, 5109-5115). This is also described

in the first subsection of the supplementary text (Sup. Info. Lines 15-18, page 2, “*Stabilization during acclimation phase*”). A similar amount of time (60 d) was employed for the experimental phase, for both low and high organic loading reactors, which was long enough to be comparable with the acclimation phase.

The recovery period was set to 14 d to evaluate if community function, structure and assembly would display signs of recovery. This was compared to the 10 d it took for the nitrite oxidation function to be reduced under press disturbance for the high organic loading reactors (Fig. S4). The amount of time allowed for recovery was indeed enough to see the complete functional recovery in two of the three previously disturbed reactors in terms of nitrite oxidation, as shown in Fig. 4 of the main manuscript. We agree with the reviewer that a longer recovery time would have provided even more information with regards to the resilience or recovery dynamics of the microbial community. Yet, the data presented in this manuscript support the claim that a particular function could recover faster than the overall community, which is what is stated in lines 204-209 (page 9): “*Additionally, two of the disturbed reactors recovered the nitrite oxidation function after returning to low organic loading conditions for 14 days (Fig. 5). However, the α - and β -diversity with respect to either taxonomy or functional genes did not revert to previous levels (Figs. 1 and 2). The fact that an altered community was still able to provide the functions of the original one (carbon removal and nitrite oxidation) supports the notion of functional redundancy⁵⁹ in complex microbial systems like bioreactors, where different types of organisms are capable of performing a wide range of functions⁶⁰.*”

To summarize, our answers to this point are already reflected in the text and no new wording has been added.

Rev. 1 – Minor Point 3: Please define acronyms in the text and not only figure headings.

Response: The revised manuscript now includes definitions for acronyms that were previously not defined like: IP2G (*InterPro to gene ontologies*, line 119, page 6), PERMANOVA (permutational analysis of variance, line 389, page 16) and PERMDISP (permutational analysis of dispersion, line 390, page 16).

Rev. 1 – Minor Point 4: i.e. should have a comma after it (i.e.,).

Response: We have now corrected it across the four times “i.e.” is mentioned on the manuscript. We thank the reviewer for pointing this out.

Rev. 1 – Minor Point 5: Please use the labelling scheme on figure S1 for Figure 1 (i.e., labels on the right side).

Response: We have made this correction on Fig. 1 following the reviewer’s suggestion.

Rev. 1 – Minor Point 6: Please define IP2G. A simple sentence will suffice.

Response: The IP2G acronym is now defined in line 119 (page 6), where it first appears in the results section: “...and the *InterPro to gene ontologies (IP2G) gene datasets*...” We thank the reviewer for pointing this out.

Rev. 1 – Minor Point 7: Page 8, line164-165, “high organic loading disturbance led to **inhibition** of nitrite oxidation function in these reactors...,” what is the evidence for inhibition. There is evidence that nitrite oxidation has decreased but inhibition would imply there is a

slowing or prevention of a process, reaction, or function by a particular **substance**. This may be semantic arguing but I would consider revising that statement, inhibition has different meanings biologically.

Response: We thank the reviewer for pointing this out. The revised version has this statement modified by replacing the word “inhibition” by the word “reduction” which is more appropriate in this context.

Lines 168-169 (page 8): “*High organic loading disturbance led to a reduction of the nitrite oxidation function in these reactors...*”

Reviewer #2 (major comments were split into sub-sections because they included several suggestions)

Rev. 2 – Comment 1: This work by Santillan and colleagues examines how a press (continuous) disturbance (increased organic loading) affects the bacterial community structure, assembly and function of sequencing batch reactors. The authors start with eight reactors that they first put through an acclimation phase (53 days). They then subject four of them (later three - one broke) to high organic carbon load while keeping the other four unaltered (60 days) before they revert the feeding scheme back to normal for 14 more days. They monitor bacterial community structure and assembly via 16S rRNA gene amplicon sequencing (104 samples - 13 time points) and metagenomics (48 samples - 7 time points), and community function via standard parameters such as chemical oxygen demand and various nitrogen species. The authors report different community structure, assembly and function in the disturbed reactors compared to the undisturbed ones. The taxa within the disturbed communities were more evenly distributed and, overall, disturbed and undisturbed communities clustered differently and performed differently except after the lift of the disturbance when function in the disturbed communities showed signs of reverting back to "normal". Moreover, community assembly was more deterministic in disturbed reactors and for common taxa. This is a study on an interesting topic, with a well-defined objective, a robust experimental setup and state-of-the-art bacterial community characterization methods. However, I have strong reservations regarding the analyses that the authors applied to assess community assembly. My greatest concern is that the authors' approach to study community assembly might be problematic for the specific nature of the dataset.

Response: We thank the reviewer for appreciating this study and for his/her very insightful and detailed comments and suggestions with regards to the null modelling methodology. We sincerely hope that the revised version together with the responses to specific points below address all of the reviewer's concerns with regards to the approach used to assess community assembly, which was particularly appropriate given the replicated design of this study.

Rev. 2 – Comment 2: The authors use the observed β -partition and generate a "null" distribution of the same metric to assess the degree of determinism in community assembly. My comments are based on downloading and running the actual R code for the model that I found here (from the SI of reference 21): https://figshare.com/articles/Sludge_bioreactor_microcosm_study_on_complex_bacterial_communities_and_ecosystem_function_using_3-CA_disturbance_at_varying_frequencies_for_35_days/7369964

Response: Yes, we provided this reference for a previous study on activated sludge microbial communities where the same model was employed, with the idea that readers could go there in case they wanted to access the code to run the model. We have decided to include the R-code for the null model and the data sets employed here in this submission, so the interested reader can more easily replicate our results. This is now included in the Data Availability statement of manuscript lines 468-469 (page 19) as follows: “*R-script to replicate the null model analysis and all other relevant data can be publicly accessed on FigShare (https://doi.org/10.6084/m9.figshare.12326330)*”.

Rev. 2 – Comment 3: What the code does is that it creates an observation list ("individuals" x samples) and shuffles the sample names to generate 10,000 null values of β -partition. Then the deterministic strength metric (DS) is calculated as per Line 386 in the main text (in the actual code SES is also calculated); this involves counting the number of species per sample as well as the total species across all samples (γ diversity - this of course remains the same in the simulations).

Response: A very important aspect of the null model, which was not mentioned by the reviewer's description within this comment, is that it also uses the number of individuals per species, which allows the model to account for not only composition (the number of species, or the richness within each bioreactor), but also for relative abundances (the number of individuals of each species, which is one read within the corresponding dataset). A detailed description of how this model works is given within the methods section in lines 402-409 (page 17): "*The model defines β -diversity as the β -partition ($\beta = 1 - \bar{\alpha}/\gamma$), taking into account both composition and relative abundances. To adapt it to handle microbial community data, we considered 'species' in the model as ASVs, genera and genes at the lowest IP2G gene level, while each individual count was one read within the corresponding dataset. The model randomizes the location of each individual within the independent replicate reactors for each of the low and high organic loading levels, while maintaining the total quantity of individuals per reactor, the relative abundance of each 'species' (i.e. ASV, genus or IP2G gene), and the γ -diversity. We applied it across different time points of the experiment.*" And also in lines 415-417 (page 17): "*Each step of the null model calculates expected mean α -diversities per treatment level and then estimates an expected β -partition. After 10,000 repetitions, the means of the distribution of random β -partitions ($\bar{\beta}_{exp}$) for each treatment level are calculated.*"

Rev. 2 – Comment 4: There are several issues with the specific metric for what the authors wish to accomplish. First, the authors state in Lines 273-274 that the metric takes into account both composition (e.g. the number of shared species like Jaccard index does) and relative abundance (e.g. like a distance metric as Bray-Curtis) but it does not.

Response: We respectfully disagree with the reviewer on this point; the null model employed in our study does account for both composition and relative abundances as stated in lines 402-404 (page 17): "*The model defines β -diversity as the β -partition ($\beta = 1 - \bar{\alpha}/\gamma$), taking into account both composition and relative abundances.*" We agree with the reviewer that relative abundances are important and should be accounted for by the null model used to assess community assembly. The null model does this by using the total number of individuals for each species (see our Response to Rev. 2 – Comment 2 above). The R-code that the reviewer found in the reference 21 states this clearly in its code-line 91: "*#This function is for use with abundance data (not presence/absence data)!*" This statement is found in Part 2A of such R-script which corresponds to the *ses.beta.function* adopted from Kraft *et al.* 2011 (DOI: 10.1126/science.1208584). For improved clarity, our revised submission includes the R-code and data used for this study via FigShare (see our "*Response to Rev. 2 – Comment 2*" above), together with the following graphical description of how the model estimates β -partitions taking into account both composition and relative abundances across different replicates. The comment: "*# This function is for use with abundance data (not presence/absence data)!*" appears in code-line 137 of the R-script we are providing via FigShare with this submission.

1) Calculation of observed β -diversity

Beta Partition
(Proportional species turnover)

$$\beta = 1 - \frac{\bar{\alpha}}{\gamma}$$

Tuomisto (2010). *Ecography*

2) Null model analysis (random shuffling of individuals across replicate bioreactors for each iteration)

Repeat for 10,000 iterations and calculate mean expected β -diversity: $\bar{\beta}_{exp}$

3) Deterministic strength calculation

$$\text{Deterministic Strength (DS)} = \left| \frac{\beta_{obs} - \bar{\beta}_{exp}}{\beta_{obs}} \right| * 100$$

Using the null-model proposed by Kraft *et al.* (2011) *Science* (DOI: 10.1126/science.1208584)
Modified R-code available in Santillan *et al.* (2020) (DOI: 10.6084/m9.figshare.12326330)

Rev. 2 – Comment 5: By the way, in line 373 the authors' reference to study 43 for using the same null model is incorrect; in that study the authors randomized GeoChip matrices (normalized mean ratio of fluorescence signal) to calculate null Jaccard similarity and Raup Crick metrics.

Response: We respectfully disagree with the reviewer on this point; the reference is correct. The study in reference 43 from Zhou *et al.* 2014 (DOI: 10.1073/pnas.1324044111) uses the null model from Kraft *et al.* 2011 (DOI: 10.1126/science.1208584). There (Zhou *et al.*), Kraft *et al.* is cited as reference 19 in the main text and as reference 22 in the supplementary information. In fact, it was the work of Zhou *et al.* (2014) (particularly Fig. 3 in that work) that inspired us to use the null model developed by Kraft *et al.* (2011) in this and also our earlier

work (Santillan *et al.* 2019, DOI: 10.1038/s41522-019-0079-4), with some adaptations to the approach. We worked directly with the null model from Kraft *et al.* (2011) because it made the R-code to run null model analysis available to other researchers.

We detailed the adaptations and improvements with regards to Zhou *et al.* (2014) approach in the supplementary info of Santillan *et al.* 2019 as follows: “*The null model analysis from Kraft et al.³⁸ was previously employed for studies in groundwater microbial communities by Zhou et al.³⁹. Our application of such model differed in two ways. First, we did not calculate pair-wise β -deviations and rather used all replicates available per treatment, since pair-wise comparisons introduce autocorrelation and bias to the output of the original method³⁸. Second, we did not use Jaccard dissimilarities which are presence-absence based (and thus not reliable for microbial community analysis), but assessed the differences in relative abundances instead, as originally proposed by Kraft et al.³⁸*

Rev. 2 – Comment 6: Second, and most importantly, this null model is highly sensitive to the ratio of per-species individuals to the number of "plots" (here: bioreactors) that is very high for most of the species in 16S rRNA amplicon and shotgun metagenomics data. If you have eight bioreactors and an SV with 100 counts in total, the probability that not a single individual of that species is assigned to all eight bioreactors is dismal ($7/8^{100}$ ~approx. $1.6E-6$). In a normal 16S rRNA gene amplicon dataset (especially from low-richness biomes like bioreactors) this is the case for the majority of SVs. Therefore this results in null mean α being very close to γ and an overestimation of DS if all the SV table is given as an input, and in the opposite if only the 100th percentile ("rare" part of the community as per the authors) is given as an input. This modeling artifact can explain the results of the authors regarding community assembly for all the examined levels (16S SVs, metagenomic taxa, metagenomic genes).

Response: This is a very important comment that gave us food for thought. We agree with the reviewer in that this model is sensitive to the ratio of per-species individuals to the number of "plots". Also, as the reviewer states, this ratio is indeed very high for most of the species in 16S rRNA amplicon and shotgun metagenomics data, which can result in an increased DS. We wish to point out, however, that when this null model is applied to the three datasets without any partition between common/rare fractions (Figs. 5A,D,G of the initial submission, Figs. 6A,D,G of the revised manuscript), the model sensitivity has no effect on the trends we observe between phases in each of those panels (L vs H), as all points in each of those panels are generated from the same dataset, and are therefore subjected to the same type of sensitivity bias.

Nonetheless, we do agree with the reviewer that a modelling artefact existed when applying the model to the 100th percentile rare fraction vs. the rest (common fraction). Therefore, we improved the treatment of input matrices as described in our Response to Rev. 2 – Comment 12 below. We also validated our results on the ASV dataset using a different null model approach following the reviewer’s suggestion, as described in Response to Rev. 2 – Comment 14 below.

Rev. 2 – Comment 7: Moreover here the statement that "It (DS) is the complement of the stochastic intensity (SI) metric defined previously..." is not correct, because this is the case only if observed- β is lower than expected- β (which, as explained above is usually not the case) otherwise the metrics are identical.

Response: We respectfully disagree with the reviewer, this definition is correct. In lines 419-421 (page 17) we define DS as: “*DS is equal to the absolute value of the difference between the observed (β_{obs}) and mean expected β -diversity, divided by the observed β -diversity: $DS =$*

$|\beta_{obs} - \overline{\beta_{exp}}| / \beta_{obs}$.” It is the complement of the stochastic intensity (SI) metric defined previously, which is calculated as follows: $SI = 1 - |\beta_{obs} - \overline{\beta_{exp}}| / \beta_{obs}$. The absolute value applied to the difference between β_{obs} and $\overline{\beta_{exp}}$ accounts for cases in which $\overline{\beta_{exp}} > \beta_{obs}$, like in the values we reported in Table S5 for the IP2G gene-level dataset (metagenomics sequencing) in the initial submission. To make this clearer, we have now included the definition of SI in the revised manuscript as follows:

Lines 421-422 (page 17): *“It is the complement of the stochastic intensity (SI) metric ($SI = 1 - |\beta_{obs} - \overline{\beta_{exp}}| / \beta_{obs}$) defined previously²¹”.*

Rev. 2 – Comment 8: Moreover here, equal units in terms of sampling effort have to be compared; in the original paper in plants the plots that were compared had the same area. However, here the authors compare rarefied (if I judge from the pre-print accompanying paper because it is not clear here) datasets that do not represent equal units in terms of sampling effort.

Response: We agree with the reviewer that equal units in terms of sampling effort have to be compared. Indeed, in this manuscript equal units in terms of sampling effort were compared across samples using rarefaction, which is an appropriate method as detailed in our “Response to Rev. 2 – Comment 9” below. With regards to the lack of clarity pointed by the reviewer in terms of whether rarefaction was or not done, we acknowledge that the previous two statements in the methods section in lines 331-332 (“*See⁴⁵ for further details including rarefaction curves*”) and lines 346-347 (“*See⁴⁵ for more details including rarefaction curves*”) were not enough to make it clear that rarefaction was done. For that reason, we have now included the rarefaction curves in this study as a supplementary figure Fig. S6 (see also our related “Response to Rev. 2 – Minor Comment 8” below). We have also added the following statement in the methods section:

Lines 409-414 (page 17): *“Samples from metagenomics datasets for both genera and IP2G genes were normalized to 100,000 reads (equivalent to 100,000 model individuals)²¹. In this manner, the number of total individuals to be shuffled in each model iteration was reduced, and the sampling effort across these datasets was equalized for null model comparisons across metagenomics data. Samples from the 16S rRNA gene ASV dataset were analysed at their rarefied values of 7,704 reads.”*

The discussion section was also adapted as follows:

Lines 282-290 (page 12): *“...with metagenomics data displaying higher stochasticity for genera compared to when ASVs from 16S rRNA gene metabarcoding were used. This was not an effect of assessing different levels of taxonomic resolution, as higher resolution levels were shown to be more conserved than lower ones⁵⁰, and therefore displayed assembly mechanisms that were more deterministic in nature. However, this could have been the effect of a different sampling coverage under the null model analysis, because metabarcoding renders a lower total number of reads per sample, about one order of magnitude, compared to metagenomics analysis. When focusing only on metagenomics data, functional gene assembly was found to be more stochastically driven than that of taxa at the genus level, even under a balanced sampling coverage.”*

Rev. 2 – Comment 9: If they would like they could properly do that by applying methods like those described in McMurdie & Holmes (2015) or in the recent Quantitative Microbiome Profiling paper by the Raes lab (the latter requires absolute counts with flow cytometry though).

Response: With all due respect to the reviewer, it has to be noticed that the state of the art is that there is not yet a consensus on what is the best or “proper” method to treat microbiome data. For example, in a recent paper by Rob Knight’s group (Weiss S. et al. 2017 Normalization and microbial differential abundance strategies depend upon data characteristics. *Microbiome*. 2017; 5(1): 27) it is mentioned that “rarefying is still a useful normalization technique: rarefying can more effectively mitigate the artifact of sample library size than other normalization techniques”. The paper from McMurdie & Holmes (2015) that the reviewer mentions describes a method (DESeq) which was developed for RNA-Seq data, which is a very different type of data than the one generated in our study. Quoting again from Weiss. S *et al.* 2017 “...many microbial environments are extremely variable in microbial composition, which would violate DESeq and edgeR-TMM normalization assumptions of a constant abundance of a majority of species and of a balance of increased/decreased abundance for those species that do change.” Furthermore, we have discussed this issue in person with Paul McMurdie himself during the ASM Microbe workshop in June 2019, and he stated that there was not yet a “best method” for pre-treating microbiome data, but that some kind of pre-treatment needed to be done. Therefore, we hold that the pre-treatment method of rarefaction is appropriate for the type of microbial datasets analysed in this manuscript.

Rev. 2 – Comment 10: Additionally, the 50% cutoff for DS (Lines 387-390) are not justified; why not switch to metrics based on z-scores to assess significance (like previously done with SES for example)?

Response: The 50% shading is not defined as a hard cut-off in terms of significance but rather as a “*quantitative estimation of the role of niche-based deterministic selection processes in shaping community composition*”, which is “*similar to SES*” as quoted from reference 43, Zhou *et al.* 2014 (DOI: 10.1073/pnas.1324044111). We are attaching a section of their work below to aid with the reviewer’s appraisal. The name of the term had to be modified to DS (deterministic strength) because our adaptation from the method described by Zhou *et al.* 2014 had several improvements, which we already detailed in our “*Response to Rev. 2 – Comment 5*” above. Therefore, DS (or SI) are used to help evaluate the relative role of deterministic vs. stochastic mechanisms of assembly under the null model employed. To clarify the reviewer’s concern we have rephrased the Methods section as follows:

Lines 422-426 (pages 17-18): “*Higher values of DS indicate a higher deviation of the observed β -diversity from the null β -diversity expectation, thus suggesting a stronger effect of deterministic-based mechanisms. Contrarily, lower DS values indicate a smaller difference between observed and null β -diversities, suggesting a more important role of stochastic mechanisms of assembly.*”

To further quantify the relative importance of deterministic vs. stochastic processes in shaping the groundwater community structure, similar to SES, the proportion of the difference between the observed similarity for each pairwise comparison and the null expected similarity divided by the observed similarity was calculated (Fig. 3). For convenience, such ratio is referred to as selection strength (SS) because it provides a quantitative estimation of the role of niche-based deterministic selection processes in shaping community composition. Its complement (1-SS) measures the relative importance of the compositional stochasticity in shaping biodiversity. As illustrated in Fig. 3, significant differences in the compositional stochasticity were observed for the communities across different time points (PERMANOVA: $F_{7,154} = 98.03$, $P < 0.001$). The stochastic processes contributed an average of 46.4% of the community variations at the early phase. However, the roles of stochastic processes in controlling community composition increased substantially during the middle phase, ranging from 81.3% to 92.0% (Fig. 3), further supporting that stochastic assembly dictated the succession of the groundwater communities in response to the EVO amendment. At the end of the experiment, the stochastic processes become less important (59.8%). In addition, the stochasticity for the majority (95%) of the detected individual functional genes showed significant correlations to that

Fig. 3. Dynamic changes of stochasticity during the succession of the groundwater microbial communities. The stochasticity is defined as the complement of the selection strength, a proportion of the differences between the observed total similarity and the null expected similarity divided by the total similarity.

Zhou et al.

PNAS | Published online February 18, 2014 | E841

The revised manuscript now includes the phylogenetic null model approach suggested by the reviewer to validate a subset of our results, as described in “Response to Rev. 2 – Comment 14” below. This also incorporates a SES (standard effect size) discussion of this work since β NTI (beta-nearest taxon index) is the SES of the MNTD (mean nearest taxon distance).

Rev. 2 – Comment 11: Finally, even if there were no problems with the applied metrics to assess determinism/stochasticity, adequate community coverage has to be demonstrated if statements about the “rare” taxa are to be made in order to ensure that the results are not because of non-detection issues. If the authors rarefied down to the level shown in the accompanying preprint (reference 45) this is certainly not the case for the 16S rRNA gene amplicon analysis at least.

Response to Rev. 2 – Comment 11: In the revised version of the manuscript we have addressed this point by improving the way input matrixes were treated to compare common vs. rare datasets (see our “Response to Rev. 2 – Comment 11” below) and by adjusting the rarefaction of the ASV dataset to increase the number of rare ASVs as described in our “Response to Rev. 2 – Minor Comment 8” below.

Rev. 2 – Comment 12: To address the above issues, the authors should revise the respective analyses by: (1) Treat the input matrices properly to ensure equal sampling effort and assess community coverage to judge if statements about the “rare” community part can be made.

Response: We are very grateful for this suggestion, as indeed the common and rare fractions evaluated did not account for equal sampling effort in the original submission. The revised version now has the null model applied in input matrices that have the same number of total counts for both common and rare fractions for each sample (reads or “individuals” under the model), with its description on lines 429-438 (page 18) of the methods section. This was done by: (i) dividing each sample into common (>90% accumulated) and rare (<10% accumulated) fractions; (ii) compiling these fractions into separate matrices of common and rare ASVs (or taxa, or genes); (iii) normalizing the number of counts per sample for the common matrix to equal that of the rare matrix; (iv) applying the null model on each matrix separately, both of them now having the same community coverage in terms of individuals accounted in the model. For example, if the total read counts in the sample is 7700 (like in the case of the ASV dataset), the common fraction would have 6930 reads while the rare fraction would have 770 reads per sample after filtering. Then the common fraction would have all its samples further normalized

to 770 reads per sample, making both datasets now comparable under the null model in terms of sampling effort. The aforementioned normalization of the common fractions of all three datasets employed in this study could be done without losing any community member (*i.e.* the total number of ASVs/taxa/genes involved in the common fraction remained the same), while keeping the relative abundances also unaltered. This procedure also led us to the adjustment of the cut-off between common and rare fractions from 99% - 1% in the original submission to 90% - 10% in the revised one. This allows a bigger number of reads per sample to be left in the rare fraction for what we believe is a more meaningful comparison between common and rare parts of the community (see the updated supplementary Table S3 for detailed information of γ - and α - diversity values involved in the null model analysis).

After this modification we could still observe that the role of stochastic assembly was stronger in the rare fraction compared with the common fraction for all three datasets evaluated (now clearer for the metagenomics genus dataset, see panels E vs F in Fig. 6). Deterministic mechanisms continued to be generally stronger for the H (press-disturbed) treatment versus the L (undisturbed) one. However, now the rare fraction seems to be driving the overall assembly of both taxonomic datasets (ASVs and metagenomics-genus), which differs from our prior submission. Fig. 6 is now as follows (Lines 695-704, page 29):

Several fragments within the discussion section (lines 255-294, pages 11-13) were modified to reflect changes in the interpretation of Figure 6 following the aforementioned corrections. Information in Table S3 of the supplementary information was also updated (lines 142-155, pages 15-18 of SI).

Rev. 2 – Comment 13: (2) Revise the null modelling using models that are appropriate for handling microbiome data, that assess determinism/stochasticity by taking into account shared species and relative abundances and that assess significance by using z-scores based on null distributions.

Response to Rev. 2 – Comment 13: In our earlier responses we have already clarified that the null-model we used took into account relative abundances (see “Response to Rev. 2 – Comment 3”). Shared species were also considered as the model shuffles individuals randomly across

samples, regardless of which species they belong to (see “Response to Rev. 2 – Comment 4”). We have also provided clarification about z-scores (SES) in our “Response to Rev. 2 – Comment 10” above.

Rev. 2 – Comment 14: (3) Complement their analyses using phylogenetic-based metrics. If the authors do not want to apply bNTI as in Stegen's framework they can use SES-PD or SES-MPD (see for example the nice paper by Burns et al (2016) in ISME journal). This would be very nice to do and it would add a lot of value to the study, given that the authors have shotgun metagenomics data from where they can assemble genomes and build genome-wise trees. Phylogenetically-informed analyses here are also very well justified given the nature of the communities in the bioreactors where phylogenetic conservatism is very strong.

Response: Following the reviewer's suggestion we have complemented our analysis by using the phylogenetic-based metric β NTI (beta-nearest taxon index) using *Phylocom* as in Stegen's framework (Webb, C. O., Ackerly, D. D. & Kembel, S. W. (2008) *Phylocom: software for the analysis of phylogenetic community structure and trait evolution*. *Bioinformatics*, 24: 2098-2100) in the ASVs from metabarcoding dataset. The method is described in the revised manuscript:

Lines 439-452: *“We further assessed assembly mechanisms by using an alternative phylogenetic-based null modelling approach on the metabarcoding ASV dataset. The model uses the β -mean nearest taxon distance (β MNTD)⁸⁵ which quantifies the phylogenetic distance between each ASV in one community, as a measure of the clustering of closely related ASVs. Phylogenetic relatedness of ASVs was characterized by multiple-alignment of ASV sequences using decipher R-package (v.2.14.0). The phylogenetic tree was then constructed and a GTR+G+I (Generalized time-reversible with Gamma rate variation) maximum likelihood tree was then fitted using the phangorn R-package (v.2.5.5). To quantify the degree to which β MNTD deviates from a null model expectation, ASVs and abundances were shuffled across the tips of the phylogenetic tree. After shuffling, β MNTD was recalculated to obtain a null value, and repeating the shuffling 1,000 times provided a null distribution. The difference between observed β MNTD and the mean of the null distribution was measured in units of standard deviation, which is referred to as the β -nearest taxon index (β NTI)⁶⁴. A value of $|\beta$ NTI| > 2 indicates that the observed turnover between a pair of communities is significantly deterministic, while $|\beta$ NTI| < 2 suggests stochastic assembly¹⁸. This analysis was done using the phylocom R-package⁸⁶.”*

The results of this analysis are included in Fig. 8 (which includes mean values and linear regression) and Fig. S5 (which distinguishes every replicate reactor with a different symbol) as follows:

Lines 712-718 (page 31): “**Fig. 8.** Nearest taxon index (β NTI) temporal dynamics for bacterial ASVs, derived from null model analysis. Calculated using (A) all, (B) common (90% acc. reads), and (C) rare ASVs (<10% acc. reads). Phases: A, acclimation (grey squares, $n = 4$); L, low organic loading (blue circles, $n = 4$); H, high organic loading (red triangles, $n = 3$). Lines represent linear regression fitting. Zones where stochastic processes dominate ($-2 < \beta$ NTI < 2) are shaded in grey. Vertical dashed line indicates the shift from high to low organic loading. Darker symbols represent average values at a given time point.”

Lines 107-113 (page 11 of SI): “**Fig. S5.** Nearest taxon index (β NTI) temporal dynamics for bacterial ASVs, derived from null model analysis. Calculated using (A) all, (B) common (90% acc. reads), and (C) rare ASVs (<10% acc. reads). Phases: A, acclimation (grey, $n = 4$); L, low organic loading (blue, $n = 4$); H, high organic loading (red, $n = 3$). Symbols represent different independent replicate reactors. Lines represent linear regression fitting. Zones where stochastic processes dominate ($-2 < \beta$ NTI < 2) are shaded in grey. Vertical dashed line indicates the shift from high to low organic loading.”

The β NTI values obtained after applying the phylogenetic-based null modelling approach to the ASV dataset indicate that the rare fraction has a higher influence of stochastic assembly mechanisms in comparison with the common fraction. Interestingly, this coincides with our observations from the previous null modelling (Fig. 6B-C). It also shows that the common fraction has a high deterministic assembly influence in terms of homogeneous selection (with most values β NTI < -2), which is in agreement with the high DS values (~99%) from the non-phylogenetic null model (Fig. 6B). However, the phylogenetic null model shows the common fraction having similar β NTI values to that of the overall ASV data, while the non-phylogenetic null model shows the rare fraction as having similar DS values to that of the overall ASV data. Some discrepancies from the application of different null models on the same data are expected given that the results from null model analyses are very sensitive to the

models, algorithms and diversity metrics employed. The β NTI approach has the advantage of including phylogeny in the analysis, but it does not take into account all independent replicates for a given time point to build the null model distribution like our original approach, therefore not taking full-advantage of the replicated design used in this study. Indeed, it is widely accepted that no null modelling approach is perfect or “the best”, as each has advantages and drawbacks (*e.g.* there is also neutral modelling, which infers the presence of a biological mechanism, but requires estimation of parameters from the same dataset and is thus prone to type I and II errors), but they can still help indicate patterns of community assembly like in the case of this study. Nonetheless, the fact that there are some concurrent results across these different null model approaches on a subset of data adds more value to the study, as suggested by the reviewer. We would like to point out that a thorough comparison of different null modelling approaches falls outside of the scope of this work. This is mentioned in the discussion section:

Lines 228-235 (page 10): “*Similarly, a recent study on granular biofilm reactors using one simple carbon source also reported stronger homogeneous selection for abundant taxa and higher stochastic assembly via drift for low-abundance taxa*³⁸. *This was done using the same type of null model as applied on the metabarcoding dataset in this study (Fig. 8), and has the advantage of incorporating phylogeny into the analysis*⁶⁴, *but does not take advantage of replicated designs the way the null model of Kraft et al.*⁶⁵ *does. As the results from null model analyses are very sensitive to the models, algorithms and diversity metrics employed*⁶⁶, *concordant outcomes in studies using different approaches are desired and thus require more research*²³.”

Rev. 2 - Minor Comment 1: Line 57: "In ecology..." I would start a new paragraph here.

Response: We followed the reviewer’s suggestion and a new paragraph now starts on Line 57 (page 4): “*In ecology, disturbances are believed to have direct effects on ecosystems...*”

Rev. 2 - Minor Comment 2: Lines 110-113: Figure 1 is actually a repetition of panels C,F and I of Figure S1. Why not just include the whole Figure S1 into the main text as Figure 1?

Response: We thank the reviewer for this observation, which is also related to our response to Rev. 1 – Major Point 4 above. For Figure 1, we chose to display only the 2nd order true α -diversity (²D) for two reasons: (i) For complex communities there is often a huge difference between the abundance of rare and abundant taxa, and overall changes in communities are driven by changes in the most abundant taxa (see van Dorst, J. et al. Community fingerprinting in a sequencing world. FEMS Microbiol. Ecol. (2014) 89, 316-330), which in terms of α -diversity are best described by ²D (see Haegeman, B. et al. Robust estimation of microbial diversity in theory and in practice. ISME J. (2013) 7, 1092-1101). (ii) The use of richness (first order α -diversity, ⁰D) to describe microbial communities is not reliable since it is heavily constrained by the method of measurement (again see Haegeman, B. et al. 2013), which makes the comparison of results from different sequencing techniques using this metric a non-meaningful one (see Shade, A. Diversity is the question, not the answer. ISME J. (2017) 11, 1-6). Since we would like to discourage the use of less robust α -diversity metrics ⁰D and ¹D, we are providing these only as part of the supplementary information. Following the reviewer’s observation, we have included a clarification in the legend of Figure S1 (lines 70-71, page 4 of SI) as follows: “*Panels from Fig. 1 (main text) are also included here to facilitate the interpretation of temporal dynamics across Hill numbers*”. To follow the reviewer’s suggestion of providing more reasoning for the reader on why we made such a choice, we have

now included a subsection in the supplementary text lines 33-45 (pages 1-2 of SI) as shown in our “Response to Rev. 1 – Major Point 4” above.

Rev. 2 - Minor Comment 3: Lines 114-116: This statement is not properly supported by the data presented. Statistical tests have to be applied in order to show a) Why were the polynomial fits chosen over other possible models and b) to show if there are significant differences in the trends of the (properly) fitted models through time.

Response: The statement mentioned by the reviewer was “*Disturbed reactors displayed higher α -diversity (2D) for taxonomic datasets of both sequencing techniques employed (Fig. 1A-B). In contrast, undisturbed reactors displayed the highest α -diversity of functional genes at the end of the disturbance phase (Fig. 1C).*” We believe it is supported by the data in Fig. 1 that show differences in terms of polynomial regression fitting using the *loess* function in R, including 95% confidence intervals. However, the original submission lacked the description of this method in the methods section, which is now included in the revised manuscript as follows:

Lines 393-394 (page 16): “*Local polynomial regression fitting was applied using the loess function from the ggplot2 package in R, including 95% confidence intervals.*”

With regards to (a) *loess* was chosen as it combines much of the simplicity of linear least squares regression with the flexibility of nonlinear regression. It does this by fitting simple models to localized subsets of the data to build up a function that describes the deterministic part of the variation in the data, point by point. In fact, one of the chief attractions of this method is that the data analyst is not required to specify a global function of any form to fit a model to the data, only to fit segments of the data. With regards to (b) we did not intend to show if there are significant differences in the trends of the models through time (which could be done using with GAMs, but would require a bigger number of data points). We instead employed Welch’s ANOVA for univariate testing, as indicated in line 394 (page 16) of the methods section, as well as lines 116-120 (page 6) of the results section: “*After the shift from high to low organic loading in disturbed reactors, treatments continued to vary in terms of α -diversity, with 2D being significantly different for both the amplicon sequence variant (ASV) ($P_{d124} = 0.0046$) and the InterPro to gene ontologies (IP2G) gene datasets ($P_{d124} = 0.0003$) based on Welch’s ANOVA.*”

Rev. 2 - Minor Comment 4: Lines 134-136: Please comment here on the overlap between the detected abundant genera with metagenomics and those with 16S rRNA gene amplicons.

Response: In this work, the 16S rRNA gene amplicon sequencing data was in agreement with metagenomics data in terms of general β -diversity patterns and main genera detected. Dynamics in abundances of nitrifier genera (*Nitrosomonas*, *Nitrospira*, *Nitrobacter*) using both approaches was shown in the study of the accompanying preprint (Ref. 45). We believe that the combination of these two different sequencing techniques, each of them with their advantages and disadvantages, provides stronger support to the different patterns in community structure reported in this study. Details about the differences between techniques have already been reported, this is why we provided a good recent reference (Knight *et al.* (2018). *Nat. Rev. Microb.* 7: 410-422) for the reader to further explore this within the discussion section as follows:

Lines 295-297 (page 13): “*The challenge of reconciling results from different sequencing methods has been recognized and requires further research*⁴¹.”

We feel that a more detailed discussion about differences between metagenomics and metabarcoding sequencing is beyond the scope of our manuscript, although we are well aware that there are many biases associated with each of these techniques as described in Table 1 of Knight *et al.* 2018 (Nat. Rev. Microb. 7: 410-422).

Further, following suggestions by reviewers 1 and 3, we have included a discussion about the role of *Paracoccus* and *Thauera* in this ecosystem:

Lines 184-191 (page 9): “*For example, the two main genera across low and high organic loading reactors (Fig. 3) were Thauera and Paracoccus, which are denitrifying organisms in activated sludge systems⁵¹. These genera likely benefited from their versatility in carbon substrate uptake and ability to reduce nitrite and nitrate available during the anoxic phase of the bioreactor cycle⁵². The accumulation of nitrite during the aerobic phase in high organic loading reactors seemed to have benefited Paracoccus more than Thauera. The main nitrifying genera were Nitrosomonas and Nitrospira, whose relative abundance was diminished in high organic loading reactors, coherent with a reduction in the nitrite oxidation function (details in Santillan et al.⁴⁵).*”

Rev. 2 - Minor Comment 5: Line 147: Here and throughout the text: Why mention the reactors that received high organic load as "press-disturbed" consistently? I think this confuses the readers into expecting other reactors disturbed differently. After the authors describe the type of disturbance for the first time they could refer to those reactors just as "disturbed".

Response: Following the reviewer's suggestion we have changed “press disturbed” reactors to just “disturbed” after the first time mentioning it.

Rev. 2 - Minor Comment 6: Lines 153-155: As explained in the main comment, this is probably an artifact of the applied null modeling and of the more-frequent non-detection (due to sampling effort) rates in the "rare" dataset.

Response: Again, we are very grateful for this suggestion. The revised version of the manuscript has now tackled this issue as explained above in our response to Rev. 2 – Comment 12.

Rev. 2 - Minor Comment 7: Lines 169-172: This is a rather strong statement. Evenness is not the sole driver of community functionality.

Response: Following the reviewer's suggestion, for clarity we have rephrased the original “*Since the low organic loading reactors displayed better COD removal and complete nitrification with almost no residual $\text{NH}_4^+\text{-N}$ or $\text{NO}_2^-\text{-N}$, it was expected that they would harbour more diverse communities, as more even communities were reported to have better functionality⁴⁸*” as follows:

Lines 173-176 (page 8): “*Since the low organic loading reactors displayed better COD removal and complete nitrification with almost no residual $\text{NH}_4^+\text{-N}$ or $\text{NO}_2^-\text{-N}$, it was expected that they would harbour more diverse communities, as community evenness was suggested to be a key factor in preserving the functional stability of an ecosystem⁴⁸.*”

We also point out that neither the prior nor the revised statement (both based on Ref. 48: Wittebolle *et al.* 2009, *Nature* 458: 623-626) claim that evenness is the sole driver of community functionality.

Rev. 2 - Minor Comment 8: Lines 338-341: It is not clear if rarefaction was performed prior to analyses but on the related reference (45) it is shown that it was actually performed and at a quite low depth to include a low-sequence-count sample.

Response: We thank the reviewer for this observation. The methods section of the original submission did indicate that rarefaction was performed for both 16S rRNA gene metabarcoding data (lines 331-332) and metagenomics sequencing data (lines 346-347), in both cases with the statement: “*See⁴⁵ for more details including rarefaction curves.*” The reviewer points out here and in his/her “Rev. 2 – Comment 11” above, that this rarefaction was done to match the sample with the lowest number of reads (3255) for the ASV dataset, as shown in panel A from Fig. S3 from the accompanying preprint (reference 45) also included here:

This choice was originally supported by the rarefied vs. observed number of ASVs chart in Panel B of that same figure, where the majority of points are close to the diagonal line showing that the vast majority of ASVs were kept after rarefaction. However, the samples 27 and 77 with the lowest number of counts in Fig. S3A (from Ref. 45) corresponded to the high organic loading bioreactor that failed as described in the methods section. We agree with the reviewer that such a low value cut for rarefaction has an effect on the rare fraction of ASVs that are subject to the null model. Hence we have removed the samples that did not correspond to the three reactors in our revised version and adjusted the rarefied number of counts to 7,704 reads. These rarefaction curves are now included as a supplementary figure (Fig. S6, see below). Panel B of that figure shows how the fit between observed and rarefied number of ASVs has improved from the previous version.

The methods section now states:

Lines 368-369 (page 15): “*Samples were rarefied to the lowest number of reads (7,704) in a sample after processing (Fig. S6A-B).*”

Lines 384-385 (page 16): “*Samples were rarefied to the lowest number of genus-level summarized reads (537,616) in a sample after processing (Fig. S6C-D).*”

Reviewer #3:

Rev. 3 - Comment 1: The authors used a two-pronged approach using 16S rRNA amplicon sequencing and shotgun metagenomics sequencing to evaluate the effect of press disturbances on the structure and function of bacterial community assembly in activated sludge bioreactors. To that end, replicate sets of bio-reactors were subjected to high and low organic feeding loads following an acclimation period of 53 days. The high organic load bio-reactors were subjected to a disturbance by switching them to a low organic substrate feed. Temporal dynamics observed in the two systems showed that disturbance leads to changes in community structure, function, and assembly. However, after the removal of the disturbance, community function bounced back, but the structure did not.

Response: We thank the reviewer for his/her comments and suggestions on this study.

Rev. 3 - Comment 2: Were the 16S data in agreement with 16S rRNA data extracted from metagenomics?

Response: We thank the reviewer for this question. In this work, the 16S rRNA gene amplicon sequencing data were in agreement with metagenomics data in terms of general β -diversity patterns and main genera detected. Dynamics in abundances of nitrifier genera (*Nitrosomonas*, *Nitrospira*, *Nitrobacter*) using both approaches was shown in the study of the accompanying preprint (Ref. 45). We believe that the combination of these two different sequencing techniques, each of them with their advantages and disadvantages, provides stronger support to the different patterns in community structure reported in this study. Details about the differences between techniques were already reported, this is why we have provided a good recent reference (Knight *et al.* (2018). *Nat. Rev. Microb.* 7: 410-422) for the reader to further explore the subject. We added the following text to the Discussion:

Lines 295-297 (page 13): “*The challenge of reconciling results from different sequencing methods has been recognized and requires further research*⁴¹”.

We feel that a more detailed discussion about differences between metagenomics and metabarcoding sequencing is beyond the scope of our manuscript, although we are well aware that there are many biases associated with these techniques as listed in Table 1 of Knight *et al.* 2018 (*Nat. Rev. Microb.* 7: 410-422). Metabarcoding faces the challenge of biases associated with different rRNA operon copy numbers, primer set employed, length of the amplicon, hypervariable region assessed, number of cycles employed in the PCR, databases used to assign taxonomy, etc. Metagenomics data on the other hand have shorter read-length fragments, a genome size bias, do not distinguish plasmids from genome, and use a different database to assign taxonomy, to mention some of the differences. Moreover, relative abundances depend on the other taxa being detected within each method, which can also result in different relative abundance values for the same genera with different sequencing techniques. Variability across replicates varied in some cases for different sequencing techniques, but the overall pattern is maintained. Sometimes a taxon is only detected either by metabarcoding or by metagenomics sequencing, which could be due to any of the biases listed in the previous paragraph, but it does not contradict the overall β -diversity pattern across press disturbed and undisturbed reactors. It should be noted that the current state of the art is that differences will occur when comparing results from these different sequencing techniques (Knight *et al.* 2018). What is important from this comparison is not so much the specific abundance values but the general trends across the low (undisturbed) and high (press disturbed) organic loading treatments, which are very similar for both methods.

Rev. 3 - Comment 3: Fig 2B needs to be plotted better. Maybe use light to dark color scheme instead of adjusting the brightness?

Response: Thank you for this comment. A light to dark colour scheme was indeed used to plot Fig. 2B, and this can be noticed when observing the black-and-white version, where colours change from light to dark grey. For the reviewer's reference, we have included the B&W version of Fig. 2B next to the original one below. No further changes were made to this figure.

Rev. 3 - Comment 4: Line 175- *Paracoccus* and *Thauera* seem to dominate in both conditions, as well as in the disturbed reactors post-disturbance. It might be a good idea to discuss what their role is in this ecosystem.

Response: In response to reviewer comments we have included a discussion about the role of *Paracoccus* and *Thauera* in this ecosystem.

Lines 184-191 (page 9): “For example, the two main genera across low and high organic loading reactors (Fig. 3) were *Thauera* and *Paracoccus*, which are denitrifying organisms in activated sludge systems⁵¹. These genera likely benefited from their versatility in carbon substrate uptake and ability to reduce nitrite and nitrate available during the anoxic phase of the bioreactor cycle⁵². The accumulation of nitrite during the aerobic phase in high organic loading reactors seemed to have benefited *Paracoccus* more than *Thauera*. The main nitrifying genera were *Nitrosomonas* and *Nitrospira*, whose relative abundance was diminished in high organic loading reactors, coherent with a reduction in the nitrite oxidation function (details in Santillan et al.⁴⁵).”

Rev. 3 - Comment 5: Line 183-186 It might be a good idea to create a heatmap of some of the key functional genes from all the reactors (similar to Fig. 3) to get a snapshot of the functional potential of the reactors and how it compares and contrasts between conditions.

Response: Thank you for this comment. Following the reviewer's suggestion (which is also related to “Rev. 1 - Comment 4” above), the revised version of the manuscript the revised version of the manuscript includes a heat map showing successional clusters of the 19 most abundant functional gene categories from the COG database (>10,000 reads) in reactors over time. This is now shown in Fig. 4 in lines 680-687 (page 27) of the manuscript:

Fig. 4. Successional clusters of the 19 most abundant functional gene categories from the COG database (>10,000 reads) across reactors over time. Z-scores denote how many standard deviations of the mean each sample contains (assigned reads per gene category, across all samples). Column legend represents day number and replicate reactors. Phases: A, acclimation ($n = 4$); L, low organic loading ($n = 4$); H, high organic loading ($n = 3$). Rectangles highlight groups of functional gene categories prevailing at different phases. Dashed vertical lines separate time points within the same phase. Dashed horizontal lines separate the four biggest clusters of trait complexes.

The discussion about changes in key functional genes across reactors was also expanded (which is also supported by supplementary Figures S3A-C) as follows:

Lines 192-203 (page 9): “We further evaluated the succession of trait-complexes, which are a product of the expression of multiple true traits⁵³, to identify patterns suggesting differential functional gene investment at different stages of the study. Traits like cell motility and cell wall were enriched in low organic loading reactors, while traits of replication and repair prevailed in high organic loading ones (Fig. 4). The enrichment of maintenance functions across disturbed reactors exemplifies how organisms have to invest resources to adapt to changes in the environment⁵⁴. High organic loading reactors also showed an increased prevalence of ATP-binding cassette transporter (Figs. S3A-B) and stress response (Fig. S3C) genes, which encode traits related to cell survival. The prevalence of certain functional genes suggests community-level tradeoffs under disturbance similar to the ones described by life-history strategy theory⁵⁵, comparable to those reported for sludge bioreactors under pollutant disturbance⁵⁶. Taken together, these observations highlight how organisms face tradeoffs when allocating resources to certain traits to maximize their fitness, which depend on abiotic and biotic interactions within their habitat^{57,58}.”

June 24, 2020

Prof. Stefan Wuertz
University of California, Davis
Civil & Environmental Engineering
One Shields Avenue
Davis, CA 95616

Re: mSystems00471-20 (Press disturbance alters community structure and assembly mechanisms of bacterial taxa and functional genes in mesocosm-scale bioreactors)

Dear Prof. Stefan Wuertz:

Overall, the reviewers were satisfied with your revisions. However, there were a few minor concerns that still need to be addressed. Once you have addressed these remaining concerns, your manuscript should be acceptable for publication in mSystems.

Below you will find the comments of the reviewers.

To submit your modified manuscript, log onto the eJP submission site at <https://msystems.msubmit.net/cgi-bin/main.plex>. If you cannot remember your password, click the "Can't remember your password?" link and follow the instructions on the screen. Go to Author Tasks and click the appropriate manuscript title to begin the resubmission process. The information that you entered when you first submitted the paper will be displayed. Please update the information as necessary. Provide (1) point-by-point responses to the issues raised by the reviewers as file type "Response to Reviewers," not in your cover letter, and (2) a PDF file that indicates the changes from the original submission (by highlighting or underlining the changes) as file type "Marked Up Manuscript - For Review Only."

Due to the SARS-CoV-2 pandemic, our typical 60 day deadline for revisions will not be applied. I hope that you will be able to submit a revised manuscript soon, but want to reassure you that the journal will be flexible in terms of timing, particularly if experimental revisions are needed. When you are ready to resubmit, please know that our staff and Editors are working remotely and handling submissions without delay. If you do not wish to modify the manuscript and prefer to submit it to another journal, please notify me of your decision immediately so that the manuscript may be formally withdrawn from consideration by mSystems.

To avoid unnecessary delay in publication should your modified manuscript be accepted, it is important that all elements you upload meet the technical requirements for production. I strongly recommend that you check your digital images using the Rapid Inspector tool at <http://rapidinspector.cadmus.com/RapidInspector/zmw/>.

Sincerely,

Sean Gibbons

Editor, mSystems

Journals Department
Reviewer comments:

Reviewer #1 (Comments for the Author):

Summary:

The revised manuscript and rebuttal by Santillan et al. has done an excellent job of addressing all reviewer's comments and I am pleased to recommend this manuscript for publication. They clearly thought through each comment and addressed it thoroughly through changes to the text and additional figures (i.e., new figure 4.). In addressing comments they have revised the manuscript to include background and discussion on around the ecological theories they are characterizing. I do have a few minor comments below, but I am happy to have been apart of this revision and am excited to see the results presented at future meetings.

Minor Points:

Please double-check the formatting of your figure call outs. For instance, ad spaces after commas (ex: line 145 (Fig. 6A,D)) should be (Fig. 6A, D) or (Fig. 6A and D). Also line 151.

If possible, please add ggplot2 package reference, I understand not all packages can be references but if it's specifically called out in a manuscript, it's worth giving credit to the developer. Same with decipher and phangorn. Well done on phylocom!

Line 437 (i.e.) needs a comma (i.e.,). Please double-check the entire manuscript.

Typo in figure 4. Amino Acid Transport (an) Metabolism. Go over each figure again.

Figures 1 and 5, for the d in Time (d), they are the only figures with the d bolded. Consistency makes for a nice manuscript.

Fig S3A add a comma to (>10000 reads)

Is the heading for Fig. S6, correct. I believe the rarefaction curves are A and C? Are you calling all the plots refraction plots?

Reviewer #2 (Comments for the Author):

In this revised version of the manuscript, the authors have addressed my main reservations about the applied model and about their interpretations of its results.

There are just some remaining points that should be made abundantly clear by the authors:

1) -Related to my previous comments 3, 4 and 5- The authors misunderstood my wording here. I specifically wrote that "**the metric** does not take into account composition and relative abundance..." whereas they interpreted this statement to be applying for **the model**. **They have kept the related confusing sentence in the manuscript (Lines 408-410) that needs to be revised. This confusion also applies to the authors' response to my comment 5; again here the shuffling model was the same but it was used for the calculation of a different metric. This should be made also clear here, despite being written in the previous authors' paper in 2019, in a brief statement stating the related modifications and referencing the 2019 work.**

2) -Related to my previous comment 6- The authors have mainly addressed my concerns about the metric's suitability for microbial sequencing datasets (although formally this should be performed with sensitivity analyses where the ratio of per-species-individuals to bioreactors is gradually changed - but this probably goes too far for this manuscript), but they have not totally disclosed the related issues of the metric in the manuscript. In the respective section (Lines 440-442) it is stated that this was performed "To ensure adequate community coverage in terms of a balanced sampling effort across both fractions prior to null model analysis..." Instead, it should be precisely written that this was performed to account for potential biases of the metric and the null model where species with many "individuals" (reads) tend to always be distributed in all "plots" (bioreactors) when the number of the latter is small compared o the number of individuals.

3) A metric of coverage such as the Good's index or equivalent still remains to be provided (in the SI) in order to ensure no major non-detection problems exist for "rare" taxa.

4) Since the authors performed β NTI analyses, a phylogenetic signal analysis in the form of a Mantel correlogram should be performed to justify the use of MNTD over MPD. Please see the related methodology in references 18 and 64. Environmental niches can be constructed easily from any relevant measured environmental parameter such as the NO₂-N concentration.

Revised submission of mSystems00471-20

Hereby we address point-by-point the comments from the three referees in the exact order they appear in the original referee report.

Reviewer #1:

Rev. 1 - Comment 1: The revised manuscript and rebuttal by Santillan *et al.* has done an excellent job of addressing all reviewer's comments and I am pleased to recommend this manuscript for publication. They clearly thought through each comment and addressed it thoroughly through changes to the text and additional figures (*i.e.*, new Figure 4). In addressing comments they have revised the manuscript to include background and discussion on around the ecological theories they are characterizing. I do have a few minor comments below, but I am happy to have been a part of this revision and am excited to see the results presented at future meetings.

Response: We thank the reviewer for his/her advice and time invested into reviewing this manuscript. Detailed responses to his/her minor comments are given below.

Rev. 1 – Minor Point 1: Please double-check the formatting of your figure call outs. For instance, add spaces after commas (ex: line 145 (Fig. 6A,D)) should be (Fig. 6A, D) or (Fig. 6A and D). Also line 151.

Response: Following the reviewer's suggestion, spaces were added after commas throughout the text for all figure call outs where needed in lines: 145 (Fig. 6A, D), 151 (Fig. 6B, E, H), 152 (Fig. 6C, F, I), 261 (Fig. 6E, H) and 262 (Fig. 6F, I).

Rev. 1 – Minor Point 2: If possible, please add *ggplot2* package reference, I understand not all packages can be referenced but if it's specifically called out in a manuscript, it's worth giving credit to the developer. Same with *decipher* and *phangorn*. Well done on *phylocom*!

Response: Following the reviewer's suggestion we have included the references to the *ggplot2* (Ref. #85), *decipher* (Ref. #87) and *phangorn* (Ref. #88) R-packages.

Rev. 1 – Minor Point 3: Line 437 (*i.e.*) needs a comma (*i.e.*). Please double-check the entire manuscript.

Response: We have added the missing comma to "*i.e.*" in Line 441. After double-checking the entire manuscript, we have also italicized "*i.e.*" in Line 270.

Rev. 1 – Minor Point 4: Typo in figure 4. Amino Acid Transport (an) Metabolism. Go over each figure again.

Response: We have now corrected the aforementioned typo in Figure 4. After double-checking the entire figure set, we have found no additional typos.

Rev. 1 – Minor Point 5: Figures 1 and 7, for the d in Time (d), they are the only figures with the d bolded. Consistency makes for a nice manuscript.

Response: For consistency, we have un-bolded (d) in Figures 1 and 7. After double-checking the entire figure set, we have also un-bolded (d) in Figure S1.

Rev. 1 – Minor Point 6: Fig S3A add a comma to (>10000 reads).

Response: We have added the aforementioned comma to (>10,000 reads) in Figure S3 legend.

Rev. 1 – Minor Point 7: Is the heading for Fig. S6, correct. I believe the rarefaction curves are A and C? Are you calling all the plots refraction plots?

Response: Indeed, we are calling all the plots in Figure S6 rarefaction plots, as all four panels include rarefaction information. In such figure, left panels (A and C) show rarefaction curves, while right panels (B and D) show rarefied versus observed number of taxa. This is stated in the legend of Figure S6 in lines 865-870 of the manuscript. However, thanks to this comment we have now found and corrected the calling for one of the left panels in line 866 that should have stated “(C)” instead of “(B)”, which might have been the source of the confusion.

Reviewer #2

Rev. 2 – Comment 1: In this revised version of the manuscript, the authors have addressed my main reservations about the applied model and about their interpretations of its results. There are just some remaining points that should be made abundantly clear by the authors.

Response: We once again thank the reviewer for his/her suggestions and time invested into reviewing this manuscript.

Rev. 2 - Minor Comment 1: (Related to my previous comments 3, 4 and 5). The authors misunderstood my wording here. I specifically wrote that "**the metric** does not take into account composition and relative abundance..." whereas they interpreted this statement to be applying for **the model**. They have kept the related confusing sentence in the manuscript (Lines 408-410) that needs to be revised. This confusion also applies to the authors' response to my comment 5; again here the shuffling model was the same **but it was used for the calculation of a different metric**. This should be made also clear here, despite being written in the previous authors' paper in 2019, in a brief statement stating the related modifications and referencing the 2019 work.

Response: We thank the reviewer for this comment, which refers to the sentence in lines 408-410 of the manuscript: "*The model defines β -diversity as the β -partition ($\beta = 1 - \bar{\alpha}/\gamma$), taking into account both composition and relative abundances.*" We now see why the statement is confusing: the β -partition metric by itself does not take into account relative abundances. It is the mean of expected β -partitions ($\overline{\beta_{\text{exp}}}$) that takes into account both composition and relative abundances, given the way it is calculated from the null-model after multiple iterations involving shuffling of individuals across plots. Therefore, following the reviewer's suggestion, we have modified the wording in the manuscript (lines 404-411) as follows:

"... *taking into account both composition and relative abundances*" has been removed. We have added lines 410-411:

"*This way it takes into account both composition and relative abundances. ... the model ...*"

Rev. 2 - Minor Comment 2: (Related to my previous comment 6). The authors have mainly addressed my concerns about the metric's suitability for microbial sequencing datasets (although formally this should be performed with sensitivity analyses where the ratio of per-species-individuals to bioreactors is gradually changed - but this probably goes too far for this manuscript), but they have not totally disclosed the related issues of the metric in the manuscript. In the respective section (Lines 440-442) it is stated that this was performed "To ensure adequate community coverage in terms of a balanced sampling effort across both fractions prior to null model analysis..." Instead, it should be precisely written that this was performed to account for potential biases of the metric and the null model where species with many "individuals" (reads) tend to always be distributed in all "plots" (bioreactors) when the number of the latter is small compared to the number of individuals.

Response: Following the reviewer's suggestion, we have modified the wording in the corresponding sub-section of the Materials and Methods portion of the manuscript, in lines 438-441 as follows: "*This accounted for potential biases in DS estimations from the null model, where taxa with many 'individuals' (reads) tend to be distributed in all 'plots'*"

(bioreactors) when the number of the latter is small compared to the number of individuals. Such normalization was done in..."

Rev. 2 - Minor Comment 3: A metric of coverage such as the Good's index or equivalent still remains to be provided (in the SI) in order to ensure no major non-detection problems exist for "rare" taxa.

Response: According to the reviewer's wishes we are providing the Chao1 index as a metric of coverage for ASV data, as well as more detailed rarefaction curves than the ones already available in Figure S6. To respect the limit of ten supplementary elements allowed by the journal, we are including this information in the FigShare depository for this manuscript, which is mentioned in the Data Availability section (<https://doi.org/10.6084/m9.figshare.12326330>).

(A) Chao1 index for ASV data, which is based on the sample coverage (the proportion of the population having representatives in the sample), plotted against observed ASV data. It assumes that the number of observations for an ASV has a Poisson distribution and corrects for variance. Here the Chao1 index is similar to the observed ASVs across all samples. (B) Rarefaction curves of ASV data per time point. Phases: A, acclimation (grey, n = 4); L, low organic loading (blue, n = 4); H, high organic loading (red, n = 3). At the rarefaction level of 7,704 reads the majority of the curves have reached a plateau. The latter coincides with the close number of observed ASVs compared to the number of rarefied ASVs reported in Figure S6B.

The Chao1 index is based on the sample coverage (the proportion of the population having representatives in the sample). It assumes that the number of observations for a taxon has a Poisson distribution and corrects for variance (Bunge, Willis and Walsh, *Annu. Rev. Stat. Appl.* 2014. 1:427–45). Panel (A) above shows that the Chao1 index is similar to the observed ASVs across all samples. Further partitioning of rarefaction curves for ASV data helps to ensure that no major non-detection problems exist for "rare" taxa. Panel (B) above shows separate rarefaction curves of samples per time point. At the rarefaction level of 7,704 reads the majority of the curves have reached a plateau. The latter coincides with the close number of observed ASVs compared to the number of rarefied ASVs reported in Figure S6B.

Rev. 2 - Minor Comment 4: Since the authors performed β NTI analyses, a phylogenetic signal analysis in the form of a Mantel correlogram should be performed to justify the use of MNTD over MPD. Please see the related methodology in references 18 and 64. Environmental niches can be constructed easily from any relevant measured environmental parameter such as the NO_2^- -N concentration.

Response: Following the reviewer's suggestion we performed a phylogenetic signal analysis in the form of a Mantel correlogram using the related methodology in references 18 and 64.

This correlogram was included as a panel D in Fig. S5, since this submission was already at the limit of ten supplementary elements allowed by the journal.

Legend of Figure S5 was modified in the manuscript, lines 860-864, to include the addition of the new panel as follows: “(D) Phylogenetic Mantel correlogram to evaluate phylogenetic signal for bacterial ASVs, relating between-ASV niche differences to between-ASV phylogenetic distances across a given phylogenetic distance. Significant Pearson correlations ($P_{FDR} < 0.05$; closed squares) suggest significant phylogenetic signal in ASV ecological niches within the associated phylogenetic distance class.”

We refer to this new panel in the results section of the revised manuscript, lines 162-164 as follows: “Phylogenetic Mantel correlogram analysis showed significant phylogenetic signal but only across relatively short phylogenetic distances (Fig. S5D), supporting the use of β NTI.”

The methods section was also modified in lines 457-463 as follows: “To support the assumption of significant phylogenetic signal, phylogenetic Mantel correlograms were constructed relating between-ASV niche differences to between-ASV phylogenetic distances across a given phylogenetic distance, following the methodology previously described^{18,64}. Environmental niches were constructed from effluent data (soluble chemical oxygen demand, ammonium, nitrite and nitrate as nitrogen). Phylogenetic distances were quantified for 50 phylogenetic distance bins and significance of Pearson correlations was evaluated using 1,000 permutations and FDR (5%) correction.”

August 3, 2020

Prof. Stefan Wuertz
University of California, Davis
Civil & Environmental Engineering
One Shields Avenue
Davis, CA 95616

Re: mSystems00471-20R1 (Press disturbance alters community structure and assembly mechanisms of bacterial taxa and functional genes in mesocosm-scale bioreactors)

Dear Prof. Stefan Wuertz:

Your manuscript has been accepted, and I am forwarding it to the ASM Journals Department for publication. For your reference, ASM Journals' address is given below. Before it can be scheduled for publication, your manuscript will be checked by the mSystems senior production editor, Ellie Ghatineh, to make sure that all elements meet the technical requirements for publication. She will contact you if anything needs to be revised before copyediting and production can begin. Otherwise, you will be notified when your proofs are ready to be viewed.

Sincerely,

Sean Gibbons
Editor, mSystems

Journals Department
Table S3: Accept
Supplemental text: Accept
Figure S6: Accept
Table S1: Accept
Figure S3: Accept
Figure S4: Accept
Figure S1: Accept
Table S2: Accept
Figure S2: Accept
Figure S5: Accept